

# Fluvial carbon dioxide emission from the Lena River basin during spring flood

**Sergey N. Vorobyev[1], Jan Karlsson[2], Yuri Y. Kolesnichenko[1], Mikhail A. Korets[3],**

**and Oleg S. Pokrovsky[4,5]***

[1]*BIO-GEO-CLIM Laboratory, Tomsk State University, Tomsk, Russia*

[2]*Climate Impacts Research Centre (CIRC), Department of Ecology and Environmental Science, Umeå University, Linnaeus väg 6, 901 87 Umeå, Sweden.*
[3] *V.N. Sukachev Institute of Forest of the Siberian Branch of Russian Academy of Sciences – separated department of the KSC SB RAS, Krasnoyarsk, 660036, Russia*
[4] *Geosciences and Environment Toulouse, UMR 5563 CNRS, 14 Avenue Edouard Belin 31400 Toulouse, France*
[5] *N. Laverov Federal Center for Integrated Arctic Research, Russian Academy of Sciences, Arkhangelsk, Russia*

Key words: $CO_2$, C, emission, permafrost, river, export, landscape, Siberia

* email: oleg.pokrovsky@get.omp.eu

**Abstract**

Greenhouse gas (GHG) emission from inland waters of permafrost-affected regions is one of the key factors of circumpolar aquatic ecosystem response to climate warming and permafrost thaw. Riverine systems of central and eastern Siberia contribute a significant part of the water and carbon (C) export to the Arctic Ocean, yet their C exchange with the atmosphere remain poorly known due to lack of *in-situ* GHG concentration and emission estimates. Here we present the results of continuous in-situ $pCO_2$ measurements over a 2600-km transect of the Lena River main stem and lower reaches of 20 major tributaries (together representing watershed area of 1,661,000 km², 66% of the Lena's basin), conducted at the peak of the spring flood. The $pCO_2$ in Lena (range 400-1400 µatm) and tributaries (range 400-1600 µatm) remained generally stable (within ca. 20 %) over the night/day period and across the river channels.

The $pCO_2$ in tributaries increased northward with mean annual temperature decrease and permafrost increase; this change was positively correlated with C stock in soil, the proportion of deciduous needle-leaf forest and the riparian vegetation. Based on gas transfer coefficients obtained from rivers of the Siberian permafrost zone ($k = 4.46$ m d$^{-1}$), we calculated $CO_2$ emission for the main stem and tributaries. Typical fluxes ranged from 1 to 2 g C m$^{-2}$ d$^{-1}$ (>99% $CO_2$, < 1 % $CH_4$) which is comparable with $CO_2$ emission measured in Kolyma, Yukon and Mackenzie and permafrost-affected rivers in western Siberia. The areal C emissions from lotic waters of the Lena watershed were quantified via taking into account the total area of permanent and seasonal water of the Lena basin (28,000 km²). Assuming 6 months of the year to be open water period with no emission under ice, the annual C emission from the whole Lena basin is estimated as $8.3 \pm 2.5$ Tg C y$^{-1}$ which is comparable to the DOC and DIC lateral export to the Arctic Ocean.

**Introduction**

Climate warming in high latitudes is anticipated to result in mobilization, decomposition and atmospheric release of significant amounts of carbon (C) stored in permafrost soils, providing a positive feedback (Schuur et al. 2015). Permafrost thawing is expected to also increase the lateral C export to rivers and lakes (Frey and Smith, 2005). The exported permafrost C is relatively labile and largely degraded to greenhouse gases (GHG) in recipient freshwaters (e.g. Vonk et al., 2015). As a result, assessment of GHG emission in rivers of permafrost affected regions is crucially important for understanding the high latitude C cycle under various climate change scenarios (Chadburn et al., 2017; Vonk et al., 2019). Among six great Arctic rivers, Lena is most emblematic one, situated chiefly within the continuous permafrost zone and exhibiting the highest seasonal variation in discharge. Over the past two decades, there has been an explosive interest to the Lena River hydrology (Yang et al., 2002; Berezovskaya et al., 2005; Smith and Pavelsky, 2008; Ye et al., 2009; Gelfan et al., 2017; Suzuki et al., 2018), organic C (OC) transport (Lara et al., 1998; Raymond et al., 2007; Semiletov et al., 2011; Goncalves-Araujo et al., 2015; Kutscher et al., 2017; Griffin et al., 2018) and general hydrochemistry

(Gordeev and Sidorov, 1993; Cauwet and Sidorov, 1996; Huh et al., 1998a,b; Huh and Edmond, 1999; Wu and Huh, 2007; Kuzmin et al., 2009; Pipko et al., 2010; Georgiadi et al., 2019; Juhls et al., 2020) including novel isotopic approaches for nutrients (Si, Sun et al., 2018) and trace metals such as Li (Murphy et al., 2019) and Fe (Hirst et al., 2020). This interest is naturally linked to the Lena River location within the forested continuous permafrost/taiga zone covered by organic-rich yedoma soil. Under on-going climate warming, the soils of the Lena River watershed are subjected to strong thawing and active (seasonally unfrozen) layer deepening (Zhang et al., 2005) accompanied by overall increase in river water discharge (McClelland et al., 2004; Ahmed et al., 2020), flood intensity and frequency (Gautier et al., 2018). The Lena River exhibits the highest DOC concentration among all great Arctic rivers (i.e., Holmes et al., 2013) which may reflect weak DOC degradation in the water column and massive mobilization of both contemporary and ancient OC to the river from the watershed (Feng et al., 2013; Wild et al., 2019). In contrast to rather limited works on $CO_2$ and $CH_4$ emissions from water surfaces of Eastern Siberia (Semiletov, 1999; Denfeld et al., 2013), extensive studies were performed on land, in the polygonal tundra of the Lena River Delta (Wille et al., 2008; Bussman, 2013; Sachs et al., 2008; Kutzbach et al., 2007) and the Indigirka Lowland (van der Molen et al., 2007). Finally, there have been several studies of sediment and particular matter transport by the Lena River to the Laptev Sea (Rachold et al., 1996; Dudarev et al., 2006) together with detailed research of the Lena River Delta (Zubrzycki et al., 2013; Siewert et al., 2016).

Surprisingly, despite such extensive research on C transport, storage, and emission in Eastern Siberian landscapes, C emissions of the Lena River main stem and tributaries remain virtually unknown, compared to a relatively good understanding of those in the Yukon (Striegl et al., 2012; Stackpoole et al., 2017), Mackenzie (Horan et al., 2019), Ob (Karlsson et al., 2021; Pipko et al., 2019) and Kolyma (Denfeld et al., 2013). The only available estimates of C emission from inland waters of the Lena basin are based on few indirect (calculated gas concentration and modelled fluxes) snapshot data with very low spatial and temporal resolution (Raymond et al., 2013). Similar to other regions, this introduces uncertainties and cannot adequately capture total regional C emissions (Abril et al., 2015; Denfeld et al., 2018; Park et al., 2018; Klaus et al., 2019; Klaus and Vachon, 2020; Karlsson et al., 2021). In particular,

no detailed studies at the peak of spring flood have been performed and the information on various
contrasting tributaries of the Lena River remains very limited. As a result, reliable estimations of
magnitude and controlling factors of C emission in the Lena River basin are poorly understood. The
present work represents a first assessment of $CO_2$ and $CH_4$ concentration and fluxes of the main stem
and tributaries during the peak of spring flow, via calculating C emission and relating these data to river
hydrochemistry and GIS-based landscape parameters. This should allow identifying environmental
factors controlling GHG concentration and emission in the Lena River watershed in order to use this
knowledge to foresee future changes in C balance of the largest permafrost-affected Arctic river.

**2. Study Site, Materials and Methods**
*2.1. Lena River and its tributaries*

97          The sampled Lena River main stem and 20 tributaries are located along a 2600 km latitudinal

transect SW to NE and include watersheds of distinct sizes, geomorphology, permafrost extent, lithology,
climate and vegetation (**Fig. 1, S1 A; Table S1**). The total watershed area of the rivers sampled in this
work is approximately 1.66 million km², representing 66% of the entire Lena River basin. Permafrost is
mostly continuous except some patches of discontinuous and sporadic in the southern part of the Lena
basin (Brown et al., 2002). The mean annual air temperatures (MAAT) along the transect ranges from -
5 °C in the southern part of the Lena basin to -9 °C in the central part of the basin. The range of MAAT
for 20 tributaries is from -4.7 to -15.9 °C. The mean annual precipitation ranges from 350-500 mm $y^{-1}$ in
the southern and south-western part of the basin to 200-250 mm $y^{-1}$ in the central and northern parts
(Chevychelov and Bosikov, 2010). The lithology of the Siberian platform which is drained by the Lena
River is highly diverse and includes Archean and Proterozoic crystalline and metamorphic rocks, Upper
Proterozoic, Cambrian and Ordovician dolostones and limestones, volcanic rocks of Permo-Triassic age
and essentially terrigenous silicate sedimentary rocks of the Phanerozoic. Further description of the Lena
River basin landscapes, vegetation and lithology can be found elsewhere (Rachold et al., 1996; Huh et
al., 1999a, b; Pipko et al., 2010; Semiletov et al., 2011; Kutscher et al., 2017; Juhls et al., 2020).
The peak of annual discharge depends on the latitude (**Fig. 1**) and occurs in May in the south
(Ust-Kut) and in June in the middle and low reaches of the Lena River (Yakutsk, Kysyr). From May 29
to June 17, 2016, we moved downstream the Lena River by boat with an average speed of 30 km h$^{-1}$
(Gureyev, 2016). As such, we followed the progression of the spring and moved from the southwest to
the northeast, thus collecting river water at approximately the same stage of maximal discharge. Note
that transect sampling is a common way to assess river water chemistry in extreme environments (Huh
and Edmond, 1999; Spence and Telmer, 2005), and generally, a single sampling during high flow season
provides the best agreement with time-series estimates (Qin et al., 2006). Regular stops each 80-100 km
along the Lena River allowed sampling for major hydrochemical parameters and $CH_4$ along the main
stem. We also moved 500-1500 m upstream of selected tributaries to record $CO_2$ concentrations for at
least 1 h and to sample for river hydrochemistry; see examples of spatial coverage in **Fig. S1 B**. From
late afternoon/evening to the next morning, we stopped for sleep but continued to record $pCO_2$ in the
Lena River main stem (15 sites, evenly distributed over the full 2600 km transect) and two tributaries
(Aldan and Tuolba).

*2.2. $CO_2$ and $CH_4$ concentrations*
Surface water $CO_2$ concentration was measured continuously, *in-situ* by deploying a portable
infrared gas analyzer (IRGA, GMT222 CARBOCAP® probe, Vaisala®; accuracy ± 1.5%) of two ranges
(2 000 and 10 000 ppm). This system was mounted on a small boat in a perforated steel pipe ~0.5 m
below water surface. The tube had two necessary opening of different diameter, which allowed free water
flow with a constant rate during the moving of the boat. The probe was enclosed within a waterproof and
gas-permeable membrane. The key to aqueous deployment of the IRGA sensor is the use of a protective
expanded polytetrafluoroethylene (PTFE) tube or sleeve that is highly permeable to $CO_2$ but
impermeable to water (Johnson et al., 2009). The material is available for purchase as a flexible tube that
fits over the IRGA sensor (Product number 200-07; International Polymer Engineering, Tempe, Arizona,
USA). We also used a copper mesh screen to minimize biofouling effects (i.e., Yoon et al., 2016).
However these effects are expected to be low in cold waters of the virtually pristine Lena River and its

tributaries. During sampling, the sensor was left to equilibrate in the water for 10 minutes before measurements were recorded.

The probe was enclosed and placed into a tube which was submerged 0.5 m below the water surface. Within this tube, we designed a special chamber that allowed low-turbulent water flow around the probe without gas bubbles. Previous studies (Park et al., 2021; Crawford et al. 2015; Yoon et al., 2016) reported some effects of boat speed on sensor $CO_2$ measurements due to turbulences. Although the turbulences were minimized in the tube/chamber design used in the present study, on a selected river transect (~10 km) we have also tested the impact of the boat speed (5, 10, 20, 30 and 40 km h$^{-1}$) on the sensor performance and have not detected any sizable (> 10%, $p < 0.05$, n = 25) difference in the $CO_2$ concentrations recorded by our system.

A Campbell logger was connected to the system allowing continuous recording of the $CO_2$ concentration (ppm), water temperature (°C) and pressure (mbar) every minute during 5 minutes over 10 minute intervals yielding 4,285 individual $pCO_2$, water temperature and pressure measurements in total. These data were averaged for 3 consecutive slots of 5 min measurements, which represented the approximate 20-km interval of the main stem route. $CO_2$ concentrations in the Lena River tributaries were measured over the first 500-2000 m distance upstream of the tributary mouth, and comprised between 5 and 34 measurements for day-time visits and between 305 and 323 individual pCO$_2$ readings for each tributary for day-time and night-time monitoring.

Sensor preparation was conducted in the lab following the method described by Johnson et al. (2009). The measurement unit (MI70, Vaisala®; accuracy $\pm 0.2\%$) was connected to the sensor allowing instantaneous readings of $pCO_2$. The sensors were calibrated in the lab against standard gas mixtures (0, 800, 3 000, 8 000 ppm; linear regression with $R^2 > 0.99$) before and after the field campaign. The sensors' drift was 0.03-0.06% per day and overall error was 4-8% (relative standard deviation, RSD). Following calibration, post-measurement correction of the sensor output induced by changes in water temperature and barometric pressure was done by applying empirically derived coefficients following Johnson et al. (2009). These corrections never exceeded 5% of the measured values. Furthermore, we tested two different sensors in several sites of the river transect: a main probe used for continuous measurements

and another probe used as a control and never employed for continuous measurements. We did not find any sizable (>10%) difference in measured $CO_2$ concentration between these two probes.

For $CH_4$ analyses, unfiltered water was sampled in 60-mL Serum bottles and closed without air bubbles using vinyl stoppers and aluminum caps and immediately poisoned by adding 0.2 mL of saturated $HgCl_2$ via a two-way needle system. In the laboratory, a headspace was created by displacing approx. 40% of water with $N_2$ (99.999%). Two 0.5-mL replicates of the equilibrated headspace were analyzed for their concentrations of $CH_4$, using a Bruker GC-456 gas chromatograph (GC) equipped with flame ionization and thermal conductivity detectors. After every 10 samples, a calibration of the detectors was performed using Air Liquid gas standards (i.e. 145 ppmv). Duplicate injection of the samples showed that results were reproducible within ±5%. The specific gas solubility for $CH_4$ (Yamamoto et al., 1976) was used in calculation of total $CH_4$ content in the vials and then recalculated to $\mu mol\ L^{-1}$ of the initial waters.

*2.3. Chemical analyses of the river water*

The dissolved oxygen (CellOx 325; accuracy of ±5%), specific conductivity (TetraCon 325; ±1.5%), and water temperature (±0.2 °C) were measured in-situ at 20 cm depth using a WTW 3320 Multimeter. The pH was measured using portable Hanna instrument via combined Schott glass electrode calibrated with NIST buffer solutions (4.01, 6.86 and 9.18 at 25°C), with an uncertainty of 0.01 pH units. The temperature of buffer solutions was within ± 5°C of that of the river water. The water was sampled in pre-cleaned polypropylene bottle from 20-30 cm depth in the middle of the river and immediately filtered through disposable single-use sterile Sartorius filter units (0.45 µm pore size). The first 50 mL of filtrate was discarded. The DOC and Dissolved Inorganic Carbon (DIC) were determined by a Shimadzu TOC-VSCN Analyzer (Kyoto, Japan) with an uncertainty of 3% and a detection limit of 0.1 mg/L. Blanks of MilliQ water passed through the filters demonstrated negligible release of DOC from the filter material.

*2.4. Flux calculation*

$CO_2$ flux ($F_{CO_2}$) was calculated following Cai and Wang (1998):

$$F_{CO_2} = K_h\, k_{CO_2}\, (C_{water} - C_{air}),\qquad\qquad(1)$$

where $K_h$ is the Henry's constant corrected for temperature and pressure (mol L$^{-1}$ atm$^{-1}$), $k_{CO_2}$ is the gas exchange velocity at a given temperature, $C_{water}$ is the water $CO_2$ concentration, and $C_{air}$ is the $CO_2$ concentration in the ambient air. In order to convert $CO_2$ concentration in water and air into $CO_2$ partial pressure, we followed Wannikhof et al. (1992) and Lauerwald et al. (2015). We used the average $CO_2$ concentrations of 402 ppm in May-June 2016 (from 129 stations all over the world, https://community.wmo.int/wmo-greenhouse-gas-bulletins), which is consistent with the value recorded at the nearest Tiksi station in 2016 (404±0.9 ppm, Ivakhov e al., 2019). Temperature-specific solubility coefficients were used to calculate respective $CO_2$ concentrations in the water following Wanninkhof et al. (1992). To standardize $k_{CO_2}$ to a Schmidt number of 600, we used the following equation (Alin et al., 2011; Vachon et al., 2010):

$$k_{600} = k_{CO_2} \left(\frac{600}{Sc_{CO_2}}\right)^{-n}\qquad\qquad(2)$$

where $Sc_{CO_2}$ is $CO_2$ Schmidt number for a given temperature (*t*, °C) in the freshwater (Wannikhof, 1992):

$$Sc_{CO_2} = 1911.1 - 118.11t + 3.4527t^2 - 0.041320t^3\qquad\qquad(3)$$

The exponent $n$ (Eqn. 2) is a coefficient that describes water surface (2/3 for a smooth water surface regime while 1/2 for a rippled and a turbulent one), and the Schmidt number for 20°C in freshwater is 600. We used $n = 2/3$ because all water surfaces of sampled rivers were considered flat and had a laminar flow (Alin et al., 2011; Jähne et al., 1987) with wind speed always below 3.7 m s$^{-1}$ (Guérin et al., 2007).

In this study, we used a $k_{CO_2}$ (a median gas transfer coefficient) value of 4.464 m d$^{-1}$ measured in the 4 largest rivers of Western Siberia Lowalnd (WSL) in June 2015 (Ob', Pur, Pyakupur and Taz rivers, Karlsson et al., 2021). These rivers are similar to Lena and its tributaries in size, but exhibit lower velocity than those of the Lena River. In fact, due to more mountainous relief, the Lena River main stem and tributaries present much higher turbulence than that of the Ob River and tributaries and as such the value

$k_{CO_2}$ used in this study can be considered rather conservative. This value is consistent with the $k_{CO_2}$ reported for the Kolyma River and its large tributaries ($3.9 \pm 2.5$ m d$^{-1}$, Denfeld et al., 2013), tributaries and main stem of the Yukon river basin ($4.9 - 7.6$ m d$^{-1}$, Striegl et al. 2012), large rivers in the Amazon and Mekong basins ($3.5 \pm 2.1$ m d$^{-1}$, Alin et al., 2011) and with modelling results of $k$ for large rivers across the world ($3 - 4$ m d$^{-1}$, Raymond et al., 2013). Note that decreasing the $k$ to most conservative value of 3 m d$^{-1}$ of Raymond et al. (2013) will decrease specific emissions by ca. 30 %.

Instantaneous diffusive CH$_4$ fluxes were calculated using an equation similar to 1 with $k$ from western Siberia rivers (Serikova et al., 2018), concentrations of dissolved CH$_4$ in the water and air–water equilibrium pCH$_4$ concentration of 1.8 ppm, and mean annual pCH$_4$ concentration in the air for 2016 (Mauna Loa Observatory fttp://aftp.cmdl.noaa.gov/products/trends/ch4/ch4_annmean_gl.txt) following standard procedures (Serikova et al., 2018, 2019).

*2.5. Landscape parameters and water surface area of the Lena basin*

The physio-geographical characteristics of the 20 Lena tributaries sampled in this study and the two points of the Lena main stem (upstream and downstream r. Aldan, **Table S1**) were determined by applying available digital elevation model (DEM GMTED2010), soil, vegetation, lithological, and geocryological maps. The landscape parameters were typified using TerraNorte Database of Land Cover of Russia (Bartalev et al., 2020, http://terranorte.iki.rssi.ru). This included various type of forest (evergreen, deciduous, needleleaf/broadleaf), grassland, tundra, wetlands, water bodies and other area. The climate and permafrost parameters of the watershed were obtained from CRU grids data (1950-2016) (Harris et al., 2014) and NCSCD data (doi:10.5879/ecds/00000001, Hugelius et al., 2013), respectively, whereas the biomass and soil OC content were obtained from BIOMASAR2 (Santoro et al., 2010) and NCSCD databases. The lithology layer was taken from GIS version of Geological map of the Russian Federation (scale 1 : 5 000 000, http://www.geolkarta.ru/). To test the effect of carbonate rocks on dissolved C parameters, we distinguished acidic crystalline, terrigenous silicate rocks and dolostones and limestones of upper Proterozoic, Cambrian and Ordovician age. We quantified river water surface area using the global SDG database with 30 m² resolution (Pekel et al., 2016) including both seasonal and

permanent water for the open water period of 2016 and for the multiannual average (reference period
2000-2004). We also used a more recent GRWL Mask Database which incorporates first order wetted
streams (Allen and Pavelsky, 2018).

248        The Pearson rank order correlation coefficient (Rs, $p < 0.05$) was used to determine the

relationship between $CO_2$ concentrations and climatic and landscape parameters of the Lena River
tributaries. Further statistical treatment of $CO_2$, DIC and DOC concentration drivers in river waters
included a Principal Component Analysis which allowed to test the effect of various hydrochemical and
climatic parameters on dissolved C pattern. For the PCA treatment, all variables were normalized as
necessary in the standard package of STATISTICA-7 (http://www.statsoft.com) because the units of
measurement for various components were different. The factors were identified via the Raw Data
method. To run the scree test, we plotted the eigen values in descending order of their magnitude against
their factor numbers. There was significant decrease in the PCA values between F1 and F2 suggesting
that a maximum of two factors were interpretable.

**3. Results**
*3.1. $CO_2$, $CH_4$, DIC and DOC in the main stem and Lena tributaries and C emission fluxes*

261        The main hydrological C parameters of the Lena River and its tributaries ($pCO_2$, $CH_4$, pH, DIC,

and DOC) are listed in **Tables 1 and 2**. Continuous $pCO_2$ measurements in the main stem (4285
individual data points) averaged for each 20 km interval over the full distance of the boat route
demonstrated a sizable increase (from ca. 380 to 1040 µatm) in $pCO_2$ northward (**Fig. 2**). There was a
positive correlation between the $pCO_2$ and distance from the head waters of the Lena River ($r = 0.625$, p
$< 0.01$, **Fig. 3 A**). The $CH_4$ concentration was low ($0.054 \pm 0.023$ and $0.061 \pm 0.028$ µmol $L^{-1}$ in the Lena
River and 20 tributaries, respectively) and did not change appreciably along the main stem and among
the 20 tributaries (**Fig. 3 B**). The DOC concentration did not demonstrate any systematic variations over
the main stem ($10.5 \pm 2.4$ mg $L^{-1}$, **Fig. 3 C**), however it was higher and more variable in tributaries (15.8
$\pm 8.6$ mg $L^{-1}$). The DIC concentration decreased about five-fold from the head waters to the middle course
of the Lena River (**Fig. 3 D**), and pH decreased by 0.8 units downstream (**Fig. 3 E**).

Generally, the concentrations of DOC measured in the present study during the peak of the spring flood are at the highest range of previous assessments during summer baseflow (around 5 mg L$^{-1}$; range of 2 to 12 mg L$^{-1}$, Cauwet and Sidorov, 1996; Lara et al., 1998; Lobbes et al., 2000; Kuzmin et al., 2009; Kutscher et al., 2017). The DIC concentration in the main stem during spring flood was generally lower than that reported during summer baseflow (around 10 mg L$^{-1}$; range of 5 to 50 mg L$^{-1}$) but consistent with values reported in Yakutsk during May and June period (7 to 20 mg L$^{-1}$, Sun et al., 2018). A sizable decrease in DIC concentration between the headwaters (first 500 km of the river) and the Lena River middle course was also consistent with the alkalinity pattern reported in previous works during summer baseflow (Pipko et al., 2010; Semiletov et al., 2011). For the Lena river tributaries, the most comprehensive data set on major ions was acquired in July-August of 1991-1996 by Huh and Edmond's group (Huh and Edmond, 1999; Huh et al., 1998a, b) and by Sun et al. (2018) in July 2012 and at the end of June 2013. For most tributaries, the concentration of DIC was a factor of 2 to 5 lower during spring flood compared to summer baseflow. This result can be explained by the strong dilution of carbonate-rich groundwaters feeding the river in spring high flow compared to summer low flow.

The measured pCO$_2$ in the river water and published (Karlsson et al., 2021) gas transfer coefficient (4.46 m d$^{-1}$) allowed for calculation of the CO$_2$ fluxes over the full length of the studied main stem (2600 km) and the sampled tributaries. Calculated CO$_2$ fluxes of the main stem and tributaries ranged from zero and slightly negative (uptake) values in the most southern part of the Lena River and certain tributaries (N Katyma), to between 0.5-2.0 g C m$^{-2}$ d$^{-1}$ in the rest of the main stem and tributaries (**Tables 1, 2; Fig. 2 B**). The largest part of the Lena River main stem, 1429 km from Kirenga to Tuolba, exhibited quite stable flux of 1.1±0.2 g C m$^{-2}$ d$^{-1}$. In the last ~400 km part of the Lena River main stem studied in this work, from Tuolba to Aldan, the calculated fluxes increased to 1.7±0.08 g C m$^{-2}$ d$^{-1}$.

The river water concentrations of dissolved CH$_4$ in the tributaries and the main channel (0.059±0.006; IQR range from 0.025 to 0.199 µmol L$^{-1}$, **Table 1, 2**) did not exhibit any trend with distance from headwaters or landscape parameters of the catchments. These values are consistent with the range of CH$_4$ concentration in the low reaches of the Lena River main channel (0.03-0.085 µmol L$^{-}$

[1]; Bussman, 2013) and are 100-500 times lower than those of $CO_2$. Consequently, diffuse $CH_4$ emissions
constituted less than 1 % of total C emissions and are not discussed in further detail.

*3.2. Diurnal (night/day) pCO₂ variations and spatial variations across the river transect*
The diel continuous $CO_2$ measurements of 3 tributaries (Kirenga, Tuolba and Aldan) and 14 sites
of the Lena River main channel showed generally modest variation with diurnal range within 10 % of
the average $pCO_2$ (**Fig. 4** and **Fig. S2**). The observed variations in $pCO_2$ between day and night were not
linked to water temperature ($p > 0.05$), which did not vary more than 1-2 °C between the day and night
period.
The spatial variations of hydrochemical parameters were tested in the upper reaches of the Lena
main stem and its largest tributary - the Aldan River (**Fig. S3**). In the Lena River, over a lateral distance
of 550 m across the river bed, the $pCO_2$ and $CH_4$ concentrations were equal to 569±4.6 µatm and
0.0406±0.0074 µmol $L^{-1}$, respectively, whereas the DIC and DOC concentrations varied < 15% (n = 5).
In the Aldan River, over a 2700 m transect across the flow, the $pCO_2$ and $CH_4$ concentrations were equal
to 1035±95 µatm and 0.078±0.00894 µmol $L^{-1}$, respectively, whereas DIC and DOC varied within < 20%
(n = 4). Overall, these results supported our design of punctual (snap shot) sampling in the middle of the
river.

*3.3. Impact of catchment characteristics on pCO₂ in tributaries of the Lena River*
The $CO_2$ concentration in the Lena River main stem and tributaries increased from southwest to
northeast (**Table 1, 2; Fig. 2**), and this was reflected in a positive (R = 0.66) correlation between $CO_2$
concentration and continuous permafrost coverage and a negative (R = -0.76) correlation with MAAT
(**Table 3**). Among different landscape factors, C stock in upper 0-30 and 0-100 cm of soil, the proportion
of riparian vegetation and bare rocks, the coverage by deciduous needle-leaf forest, and coverage of river
watershed by water bodies (mostly lakes) exhibited significant ($p < 0.01$, *n* = 19) positive correlations
($0.54 \leq R \leq 0.86$) with average $pCO_2$ of the Lena River tributaries (**Fig. 5**). The other potentially important
landscape factors of the river watershed (proportion of peatland and bogs, tundra coverage, total

aboveground vegetation, type of permafrost, annual precipitation) did not significantly impact the $CO_2$ concentration in the Lena River tributaries (**Table 3**).

Further assessment of landscape factor control on C parameters of the river water was performed via a PCA. This analysis basically confirmed the results of linear regressions and revealed two factors capable explaining only 12.5 and 3.5% of variability (**Fig. S4**). The F1 strongly acted on the sample location at the Lena transect, the content of OC in soils, the watershed coverage by deciduous needle-leaf forest and shrubs, riparian vegetation (a proxy for the width of the riparian zone), but also proportion of tundra, bare rock and soils, water bodies, peatland and bogs (> 0.90 loading). The $pCO_2$ was significantly linked to F1 (0.72 loading).

*3.4. Areal emission from the Lena River basin*

The areal emission of $CO_2$ from the lotic waters of the Lena River watershed were assessed based on total river water coverage of the Lena basin in 2016 (28,197 km², of which 5,022 km² is seasonal water, according to the Global SDG database). This value is consistent with the total river surface area from the GRWL Mask database (22,479 km²). Given that the measurements were performed at the peak of spring flood in 2016, we used the maximal water coverage of the Lena River basin.

Based on past calculated $pCO_2$ of the Lena River (400 - 1000 µatm, Semiletov, 1999; Semiletov et al., 2011; Pipko et al., 2010) both the seasonality and spatial differences downstream are relatively small. Indeed, for the lower reaches of the Lena River, from Yakutsk to the Lena Delta, Semiletov (1999) and Semiletov et al. (2011) reported, for August-September 1995, the average $pCO_2$ of 538±96 µatm (range 380-727 µatm). This value is very similar to the one obtained in July 2003 for the low reaches of Lena (559 µatm, Pipko et al., 2010). Over the full length of the Lena River, from Ust-Kut to the Lena mouth, Pipko et al. (2010) reported an average $pCO_2$ of $450 \pm 100$ µatm in June-July 2003. At the same time, calculated $pCO_2$ from previous field campaigns are generally lower than the $pCO_2$ of the Lena River main stem directly measured in this study: 700-800 µatm for the Ust Kut – Nuya segment (1331 km); 845 – 1050 µatm for the Nuya – Aldan segment (1050 km).

Thus, despite the absolute values of calculated $pCO_2$ involving uncertainties (our calculated: measured $pCO_2$ in Lena River main channel and tributaries equaled $0.67\pm0.15$ (n = 47)), this suggests spatial and temporal stability of the $pCO_2$ in the Lena River waters and allows for extrapolation of the measured $pCO_2$ in the Lena River from Yakutsk to Aldan to the lower reaches of the river. As for the Lena tributaries, to the best of our knowledge there is no published information on $pCO_2$ concentration and emission. Overall, the major uncertainty in estimation of the Lena River basin emission stems from a lack of direct $pCO_2$ measurements in the northern part of the main channel over ca. 1000 km downstream of the Aldan River including the large tributary Vilyi. Further, we noted that the largest northern tributary, the Aldan River providing 70% of the spring time discharge of the Lena River (Pipko et al., 2010), demonstrated sizably higher emissions compared to the Lena River main channel upstream of Aldan ($3.2\pm0.5$ and $1.69\pm0.08$ g C m$^{-2}$ d$^{-1}$, respectively).

For areal emission calculations, we used the range of $CO_2$ emissions from 1 to 2 g C m$^{-2}$ d$^{-1}$ which covers full variability of both large and small tributaries and the Lena River main channel (**Tables 1-2, Figure 2 B**). This estimation assumes lack of $pCO_2$ dependence on the size of the watershed in the Lena basin as confirmed by our data (**Fig. S5**). For an alternative areal emission calculations, we explicitly took into account the water area of the main stem (43% relative to the total water area of the Lena catchment) and we introduced the partial weight of emission from the 3 largest tributaries (Aldan, Olekma and Vitim) according to their catchment surface areas (43, 12 and 14% of all sampled territory, respectively). We summed up contribution of the Lena river main stem and the tributaries and we postulated the average emission from the main stem upstream of Aldan ($1.25\pm0.30$ g C m$^{-2}$ d$^{-1}$) as representative of the whole Lena River. This resulted in an updated value of $1.65\pm0.5$ g C m$^{-2}$ d$^{-1}$ which is within the range of 1 to 2 g C m$^{-2}$ d$^{-1}$ assessed previously. Note that this value is most likely underestimated because emissions from the main stem downstream of Aldan are at least 10 % higher (Table 1, Fig. 1 B).

For the two months of maximal water flow (middle of May - middle of July), the C emission from the whole Lena basin equates to $2.8 \pm 0.85$ Tg C which is 20 to 30% of the DOC and DIC lateral export to the Arctic Ocean. Assuming six months of open water period and no emission during winter, this

yields $8.3 \pm 2.5$ Tg C y$^{-1}$ of annual emission for the whole Lena River basin (2,490, 000 km²) with a total
lotic water area of 28,100 km². Considering the only 23,200 km² water area in July-October (and maximal
water coverage in May-June), these numbers decrease by 12% which is below the uncertainties
associated with our evaluation.


**4. DISCUSSION**
*4.1. Possible driving factors of $CO_2$ pattern in the Lena River basin*
Generally, the SW to NE increase in $CO_2$ concentrations and fluxes of the tributaries was
consistent with $CO_2$ pattern in the main stem (**Fig. 2; Tables 1, 2**), and thus can be considered as a
specific feature of $CO_2$ exchange between lotic waters and atmosphere in the studied part of the Lena
Basin. At the same time, there were sizable local variations (peaks) in $CO_2$ concentration of the main
stem along the sampling route (**Fig. 2 A**). Peaks shown on the diagram of the main stem are not
necessarily linked to $CO_2$-rich tributaries, but likely reflect local processes in the main stem, including
lateral influx from the shores and shallow subsurface waters, which is typical for permafrost regions of
forested Siberian watersheds (i.e., Bagard et al., 2011). Given that the data were averaged over ~20-km
distance, we believe that these peaks are not artifacts but reflect local heterogeneity of the $pCO_2$ pattern
in the main stem (turbulences, suprapermafrost water discharge, sediment resuspension and respiration).
Note that such a heterogeneity was not observed in the tributaries, at least at the scale of our spatial
coverage (see **Fig. S1 B, S3**).
The PCA demonstrated extremely low ability to describe the data variability (12% by F1 and only
3.5% by F2). We believe that the most likely reason of weak PCA capacity is the rather homogeneous
distribution of $CO_2$ and $CH_4$ among the tributaries, primarily linked to the specific hydrological period,
studied in this work - the spring flood. During this high flow period, the local lithological and soil
heterogeneities among tributaries or the segments of the main stem virtually disappear and surface flow
(via vegetation leaching) becomes the most important driver of riverine chemistry, as is known from

adjacent permafrost territories in Central Siberia (i.e., Bagard et al., 2011). Nevertheless, some specific features of the data structure could be established. The first factor, significantly linked to $pCO_2$ (0.72 loading), strongly acted on the sample location at the Lena transect, the watershed coverage by deciduous needle-leaf forest and shrubs, riparian vegetation, and also the proportion of tundra, bare rock and soils, water bodies, peatland and bogs (> 0.90 loading). This is fully consistent with spatial variation of $pCO_2$ along the permafrost and climate gradient in the main channel and sampled tributaries. Positive loading of riparian vegetation, peatlands and bogs on F1 (0.927 and 0.989, respectively) could reflect a progressive increase in the feeding of the river basin by mire waters, an increase in the proportion of needle-lead deciduous trees, and in the width of the riparian zone from the SW to the NE direction.

Lack of sizable variation in $pCO_2$ between the day and night period or across the river bed suggests quite low site-specific and diurnal variability. It may be indicative of a negligible role of primary productivity in the water column given the low water temperatures, shallow photic layer of organic-rich and turbid waters and lack of periphyton activity during high flow of the spring flood. The $pCO_2$ increased by a factor of 2 to 4 along the permafrost/temperature gradient from the southwest to the northeast, for both the main channel and sampled tributaries. This may reflect progressive increase in the feeding of the river basin by mire waters, increase in the proportion of needle-leaf deciduous forest, and an increase in the width of the riparian zone. Another strong correlation is observed between the stock of OC in soils (both 0-30 and 0-100 cm depth) and the $pCO_2$ of Lena tributaries. Organic-rich soils are widely distributed in the central and northern part of the basin. The most southern part of the Lena basin is dominated by carbonate rocks and crystalline silicates in generally mountainous terrain, where only thin mineral soils are developed. The northern (downstream of the Olekma River) part of the basin consists of soils developed on sedimentary silicate rocks as well as vast areas of easily eroded yedoma soils. It is likely that both organic matter mineralization in OC-rich permafrost soils and lateral export of $CO_2$ from these soils, together with particulate and dissolved OC export and mineralization in the water column, are the main sources of $CO_2$ in the river water. Although some studies have demonstrated high lability of DOM in arctic waters (Cory et al., 2014; Ward et al., 2017; Cory and Kling, 2018), others suggest that DOM photo- and bio-degradation is low and does not support the major part of $CO_2$

supersaturation in water (Shirokova et al., 2019; Payandi-Rolland et al., 2020; Laurion et al., 2021). Note
that we have not observed any significant relationship between the DOC and $pCO_2$ in the Lena River and
tributaries (**Fig. S6 A**). Lack of such a correlation and absence of diurnal $pCO_2$ variations imply that in-
stream processing of dissolved terrestrial OC is not the main driver of $CO_2$ supersaturation in the river
waters of the Lena River basin. Furthermore, a lack of lateral (across the river bed) variations in $pCO_2$
does not support a sizable input of soil waters from the shore, although we admit that much higher spatial
coverage along the river shore is needed to confirm this hypothesis.

438       The role of underground water discharge in regulating $pCO_2$ pattern of the tributaries is expected

to be most pronounced in the SW part of the basin (Lena headwaters), where carbonate rocks of the
basement would provide sizable amounts of $CO_2$ discharge in the river bed. However, there was no
relationship between the proportion of carbonate rocks on the watershed and the $pCO_2$ in the tributaries
(**Fig. S6 B**). Furthermore, for the Lena River main stem, the lowest $CO_2$ concentrations were recorded in
the upper reaches (first 0-800 km) where carbonate rocks dominate. Altogether, this makes the impact of
$CO_2$ from underground carbonate reservoirs on river water $CO_2$ concentrations unlikely. This is further
illustrated by a lack of correlation between $pCO_2$ and DIC or pH (**Fig. S7 A** of the Supplement). The pH
did not control the $CO_2$ concentration in the main stem and only weakly impacted the $CO_2$ in the
tributaries (Fig. **S7 B**). The latter could reflect an increase in $pCO_2$ in the northern tributaries which
exhibited generally lower pH compared to the SW tributaries hosted within the carbonate rocks. Overall,
such low correlations of $CO_2$ with DIC and pH reflected a generally low predictive capacity to calculate
$pCO_2$ from measured pH, temperature and alkalinity (see section 3.4).

451       Therefore, other sources of riverine $CO_2$ may include particulate organic carbon processing in the

water column (Attermeyer et al., 2018), river sediments (Humborg et al., 2010) and within the riparian
zone (Leith et al., 2014, 2015) which require further investigation. In addition, although there was no
sizable variation in $pCO_2$ between the day and night period or across the river bed, the flux could show
different spatial and temporal patterns if $k$ shows larger variability (cf., Beaulieu et al., 2012). This calls
for a need of direct flux measurements in representative rivers and streams of the Lena River basin.
Overall, the present study demonstrates highly dynamic and non-equilibrium behavior of $CO_2$ in the river

waters, with possible *hot spots* from various local sources. For these reasons, *in-situ*, high spatial resolution measurements of $CO_2$ concentration in rivers—such as those reported for the Lena Basin in this study—are crucially important for quantifying the C emission balance in lotic waters of high latitudes.

*4.2. Areal emission from the Lena River basin vs lateral export to the Arctic Ocean*

The estimated $CO_2$ emissions from the Lena River main channel over 2600 km distance (0.8 – 1.7 g C $m^{-2}$ $d^{-1}$) are comparable to values directly measured in rivers and streams of the continuous permafrost zone of western Siberia (0.98 g C $m^{-2}$ $d^{-1}$, Serikova et al., 2018), the Kolyma River (0.35 g C $m^{-2}$ $d^{-1}$ in the main stem; 2.1 g C $m^{-2}$ $d^{-1}$ for lotic waters of the basin), and the Ob River main channel (1.32±0.14 g C $m^{-2}$ $d^{-1}$ in the permafrost-free zone, Karlsson et al., 2021). At the same time, the Lena River flux ($FCO_2$) values are lower than typical emissions from running waters in the contiguous Unites States (3.1 g C $m^{-2}$ $d^{-1}$, Hotchkiss et al., 2015), small mountain streams in Northern Europe (3.3 g C $m^{-2}$ $d^{-1}$, Rocher-Ros et al., 2019), and small streams of the northern Kolyma River (6 to 7 g C $m^{-2}$ $d^{-1}$, Denfeld et al., 2013) and Ob River in the permafrost-affected zone (3.8 to 5.4 g C $m^{-2}$ $d^{-1}$, Karlsson et al., 2021). In contrast to the main stem, the range of $FCO_2$ in the tributaries is larger (0.2 to 3.2 g C $m^{-2}$ $d^{-1}$) and presumably reflects a strong variability in environmental conditions across a sizable landscape and climate transect.

Total C emissions from other major Eastern Eurasian permafrost-draining rivers (i.e. sum of Kolyma, Lena and Yenisei rivers) based on indirect estimates (40 Tg C $y^{-1}$, Raymond et al., 2013) are generally supportive of the estimations of the Lena River in this study (5 to 10 Tg C $y^{-1}$). At the same time, the C emission from the Lena river basin (28,100 km² water area) are lower than those of the lotic waters of western Siberia (30 Tg C $y^{-1}$ for 33,389 km² water area, Karlsson et al., 2021). The latter drain through thick, partially frozen peatlands within the discontinuous, sporadic and permafrost-free zones, which can cause high OC input and processing and, thus, enhanced C emissions (Serikova et al., 2018).

Despite the high uncertainty on our regional estimations [due to lack of directly measured gas transfer values and low seasonal resolution] we believe that these estimations are conservative and can

be considered as first order values pending further improvements. In order to justify extrapolation of our data to all seasons and the entire area of the Lena basin, we analyzed data for spatial and temporal variations in $pCO_2$ of the Lena River main stem from available literature. From the literature there were three important findings. First, based on published data, the seasonal and spatial variabilities of $pCO_2$ across the majority of the Lena River main stem are not high during open water period, although the low reaches of the Lena River may exhibit higher emissions compared to the middle and upper course (see **section 3.4**). Second, although small mountainous headwater streams of the tributaries may exhibit high $k$ due to turbulence, this could be counteracted by lower $CO_2$ supply from low OC in mineral soil, lack of riparian zone and scarce vegetation. Third, although these small streams (watershed area < 100 km²) may represent > 60% of total watershed surfaces of the Lena basin (Ermolaev et al., 2018), their contribution to the total water surface is < 20% (19% from combined analysis of DEM GMTED2010 and 16% from the GRWL or Global SDG database as estimated in this study). Therefore, given that (*i*) within the stream-river continuum, the $CO_2$ efflux increases only two-fold demonstrating a discharge decrease by a factor of 10,000 (from 100 to 0.01 m$^3$ s$^{-1}$, Hotchkiss et al., 2015), and (*ii*) the watershed area had no impact on $pCO_2$ in the river water (**Fig. S5**), this uncertainty is likely less important. As such, instead of integrating indirect literature data, we used the $pCO_2$ values measured in the present study to calculate the overall $CO_2$ emission from all lotic waters of the Lena basin.

The C evasion from the Lena basin assessed in the present work is comparable to the total (DOC+DIC) lateral export by the Lena River to the Arctic Ocean (10 Tg C y$^{-1}$ by Semiletov et al. (2011), or 11 Tg C y$^{-1}$ (5.35 Tg DIC y$^{-1}$ + 5.71 Tg DOC y$^{-1}$ by Cooper et al. (2008)). Moreover, the C evasion strongly exceeds sedimentary C input to the Laptev Sea by all Siberian rivers (1.35 Tg C y$^{-1}$, Rachold et al. (1996) and Dudarev et al. (2006)), the Lena River annual discharge of particulate organic carbon (0.38 Tg y$^{-1}$, Semiletov et al., 2011), and OC burial on the Kara Sea Shelf (0.37 Tg C y$^{-1}$, Gebhardt et al., 2005).

Typical concentrations of $CH_4$ in the Lena tributaries and the main channel are 100 to 500 times lower than those of $CO_2$. Given that the global warming potential (GWP) of methane on a 100-year scale is only 25 times higher than that of $CO_2$, the long-term role of diffuse methane emission from the Lena River basin is still 4 to 20 times lower than that of $CO_2$. However, on a short-term scale (20 years), the

GWP of methane can be as high as 96 (Alvarez et al., 2018) and its role in climate regulation becomes comparable to that of the $CO_2$. This has to be taken into account for climate modeling of the region.

The follow up studies of this large heterogenous and important system should include $CO_2$ measurements in 1) the low reaches of the Lena River, downstream of Aldan, notably large organic-rich tributaries such as Vilyi (454,000 km²) and where the huge floodzone (20-30 km wide) with large number of lakes and wetlands is developed, and 2) highly turbulent eastern tributaries of the Lena River downstream of Aldan, which drain the Verkhoyansk Ridge and are likely to exhibit elevated gas transfer coefficients.

## 5. Conclusions

Continuous $pCO_2$ measurements over 2600 km of the upper and middle part of the Lena River main channel and 20 tributaries during the peak of spring flood allowed to quantify, for the first time, in-situ $pCO_2$ variations which ranged from 500 to 1700 µatm and exhibited a 2 to 4-fold increase in $CO_2$ concentration northward. There was no major variation in $pCO_2$ between the day and night period or across the river bed which supports the chosen sampling strategy. The northward increase in $pCO_2$ was correlated with an increased proportion of needle-leaf deciduous trees, the width of the riparian zone and the stock of organic C in soils. Among the potential drivers of riverine $pCO_2$, changes in the vegetation pattern (northward migration of larch tree line in Siberia; Kruse et al., 2019) and soil OC stock are likely to be most pronounced during ongoing climate warming and thus the established link deserves further investigation. The total C emission from the lotic waters of the Lena River basin ranges from 5 to 10 Tg C $y^{-1}$ which is comparable to the annual lateral export (50% DOC, 50 % DIC) by the Lena River to the Arctic Ocean. However, these preliminary estimations of C emission should be improved by direct flux measurements across seasons in different types of riverine systems of the basin, notably in the low reaches of the Lena River.

**Acknowledgements.**

We acknowledge support from an RSF grant 18-17-00237_P, the Belmont Forum Project VULCAR-FATE, and the Swedish Research Council (no. 2016-05275). We thank the Editor Ji-Hyung Park and 3 anonymous reviewers for their very constructive comments. Chris Benker is thanked for English editing.

**Authors contribution.**

SV and OP designed the study and wrote the paper; SV, YK and OP performed sampling, analysis and their interpretation; MK performed landscape characterization of the Lena River basin and calculated water surface area; JK provided analyses of literature data, transfer coefficients for $FCO_2$ calculations and global estimations of areal emission vs export.

**Competing interests.**

The authors declare that they have no conflict of interest.

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

**Table 1.** Measured water temperature, $pCO_2$, calculated $CO_2$ flux, $CH_4$, DOC, and DIC concentrations
and pH in the Lena River main stem (average ± s.d.; (n) is number of measurements).

| River transect | $T_{water}$, °C | $pCO_2$, µatm | $FCO_2$, g C m$^{-2}$ d$^{-1}$ $k = 4.464$ |
|---|---|---|---|
| Lena upstream of Kirenga (0-578 km) | 12.65±0.22 (99) | 714±22 (99) | 0.849±0.061 (99) |
| Lena Kirenga – Vitim (579-1132 km) | 9.17±0.15 (87) | 806±8.8 (87) | 1.19±0.024 (87) |
| Lena Vitim -Nuya (1132-1331 km) | 8.10±0.115 (27) | 797±22 (27) | 1.22±0.072 (27) |
| Lena Nuya – Tuolba (1331-2008 km) | 9.61±0.09 (95) | 846±12 (95) | 1.29±0.034 (95) |
| Lena Tuolba – Aldan (2008-2381 km) | 10.6±0.21 (52) | 1003±28 (52) | 1.69±0.081 (5) |


| | $CH_4$, µmol L$^{-1}$ | DOC, mg L$^{-1}$ | DIC, mg L$^{-1}$ | pH |
|---|---|---|---|---|
| Lena upstream of Kirenga (0-578 km) | 0.068±0.003 (6) | 13.9±1.4 (6) | 20.0±1.2 (6) | 8.12±0.203 (7) |
| Lena Kirenga – Vitim (579-1132 km) | 0.040±0.002 (12) | 7.55±0.246 (14) | 6.30±0.485 (14) | 7.77±0.040 (14) |
| Lena Vitim -Nuya (1132-1331 km) | 0.038±0.003 (5) | 9.02±0.29 (3) | 4.55±0.70 (3) | 7.69±0.063 (3) |
| Lena Nuya – Tuolba (1331-2008 km) | 0.037±0.002 (6) | 10.4±0.78 (2) | 5.09±1.157 (2) | 7.62±0.052 (2) |
| Lena Tuolba – Aldan (2008-2381 km) | 0.088±0.034 (5) | 11.6±0.27 (5) | 5.24±0.102 (5) | 7.49±0.044 (5) |



**Table 2.** Measured water temperature, $pCO_2$, calculated $CO_2$ flux, $CH_4$, DOC, DIC concentration and pH in the tributaries (average $\pm$ s.d.; (n) is number of measurements).

| Tributary | $T_{water}$, °C | $pCO_2$, µatm | $FCO_2$, g C m$^{-2}$ d$^{-1}$ |
|---|---|---|---|
| №4 Orlinga (208 km) | 8.0±0.0 (13) | 515±2.9 (13) | 0.347±0.01 (13) |
| №5 Nijnaya Kitima (228 km) | 6.8±0.0 (11) | 462±9.4 (11) | 0.193±0.03 (11) |
| №8 Taiur (416 km) | 8.5±0.0 (10) | 575±31 (10) | 0.523±0.095 (10) |
| №10 Bol. Tira (529 km) | 11.9±0.0 (15) | 788±12 (15) | 1.04±0.03 (15) |
| №12 Kirenga (579 km) | 10.2±0.0 (323) | 448±4 (323) | 0.131±0.01 (323) |
| №25 Thcayka (1025 km) | 8.6±0.01 (8) | 856±13 (8) | 1.37±0.04 (8) |
| №28 Tchuya (1110 km) | 5.9±0.0 (5) | 751±5.7 (5) | 1.16±0.019 (5) |
| №29 Vitim (1132 km) | 6.8±0.0 (10) | 654±10 (10) | 0.812±0.03 (10) |
| №32 Ykte (1265 km) | 4.9±0.0 (11) | 676±4.8 (11) | 0.943±0.02 (11) |
| №34 Kenek (1312 km) | 7.60±0.0 (11) | 710±2.6 (11) | 0.964±0.01 (11) |
| №36 Nuya (1331 km) | 11.8±0.0 (10) | 752±6.0 (10) | 0.947±0.02 (10) |
| №38 Bol. Patom (1670 km) | 6.9±0.0 (5) | 730±12 (5) | 1.05±0.04 (5) |
| №39 Biriuk (1712 km) | 14.2±0.0 (5) | 929±19 (5) | 1.32±0.05 (5) |
| №40 Olekma (1750 km) | 6.4±0.0 (11) | 802±14 (11) | 1.30±0.05 (11) |
| №43 Markha (1948 km) | 17.5±0.0 (15) | 844±15 (15) | 0.998±0.03 (15) |
| №44 Tuolba (2008 km) | 12.3±0.0 (305) | 1181±6 (305) | 2.08±0.02 (305) |
| №46 Siniaya (2118 km) | 18.5±0.0 (24) | 894±19 (24) | 1.08±0.04 (24) |
| №48 Buotama (2170 km) | 18.5±0.0 (24) | 1160±25 (24) | 1.66±0.06 (24) |
| №52-54 Aldan (2381 km) | 14.8±0.02 (316) | 1715±12 (316) | 3.23±0.03 (316) |

**Table 2,** continued.

| | CH$_4$, µmol L$^{-1}$ | DOC, mg L$^{-1}$ | DIC, mg L$^{-1}$ | pH |
|---|---|---|---|---|
| №4 Orlinga (208 km) | 0.064 | 13.4 | 27.9 | 8.64 |
| №5 Nijnaya Kitima (228 km) | 0.033 | 16.7 | 13.1 | 8.48 |
| №8 Taiur (416 km) | 0.079 | 10.0 | 11.2 | 8.36 |
| №10 Bol. Tira (529 km) | 0.084 | 22.7 | 14.9 | 8.13 |
| №12 Kirenga (579 km) | 0.036 | 5.13 | 6.86 | 7.97 |
| №25 Thcayka (1025 km) | 0.066 | 16.7 | 22.5 | 8.30 |
| №28 Tchuya (1110 km) | 0.037 | 7.08 | 3.44 | 7.57 |
| №29 Vitim (1132 km) | 0.057 | 10.1 | 2.18 | 7.70 |
| №32 Ykte (1265 km) | 0.037 | 5.49 | 15.3 | 7.86 |
| №34 Kenek (1312 km) | 0.053 | 21.1 | 16.0 | 8.12 |
| №36 Nuya (1331 km) | 0.048 | 26.6 | 11.7 | 7.80 |
| №38 Bol. Patom (1670 km) | 0.026 | 6.99 | 4.56 | 7.76 |
| №39 Biriuk (1712 km) | 0.047 | 29.2 | 11.3 | 7.87 |
| №40 Olekma (1750 km) | 0.046 | 13.3 | 3.3 | 7.53 |
| №43 Markha (1948 km) | 0.088 | 27.4 | 10.9 | 8.00 |
| №44 Tuolba (2008 km) | 0.035 | 14.5 | 14.7 | 7.98 |
| №46 Siniaya (2118 km) | 0.113 | 33.2 | 7.73 | 7.97 |
| №48 Buotama (2170 km) | 0.124 | 12.2 | 31.6 | 8.45 |
| №52-54 Aldan (2381 km) | 0.088 (4) | 9.07±0.75 (4) | 6.67±0.13 (4) | 7.59±0.02 (4) |

Footnote : in all tributaries except Aldan, there was only one measurement of CH$_4$, DOC, DIC and pH

**Table 3.** Pearson correlation coefficients (**R**) between $pCO_2$ and landscape parameters of the Lena tributaries. Significant correlations ($p < 0.05$) are marked by asterisk. Methane concentration did not exhibit any significant correlation with all tested parameters.


| % coverage of the watershed and climate | R 948 |
|---|---|
| Broadleaf Forest | 0.04 |
| Humid Grassland | -0.52* |
| Shrub Tundra | -0.05 |
| Riparian Vegetation | 0.87* |
| Croplands | -0.31 |
| Bare Soil and Rock | 0.54* |
| Evergreen Needle-leaf Forest | -0.59* |
| Deciduous Broadleaf Forest | -0.14 |
| Mixed Forest | -0.34 |
| Deciduous Needle-leaf Forest | 0.56* |
| Bogs and marches | 0.44 |
| Palsa bogs | 0.29 |
| Recent burns | -0.25 |
| Water bodies | 0.63* |
| Aboveground biomass | -0.55* |
| Soil C stock, 0-30 cm | 0.54* |
| Soil C stock, 0-100 cm | 0.65* |
| Carbonate rocks | 0.20 |
| Continuous permafrost | 0.66* |
| Discontinuous permafrost | -0.27 |
| Sporadic permafrost | -0.43 |
| Isolated permafrost | -0.19 |
| Mean annual air temperature | -0.76* |
| Mean annual precipitation, mm | 0.10 |



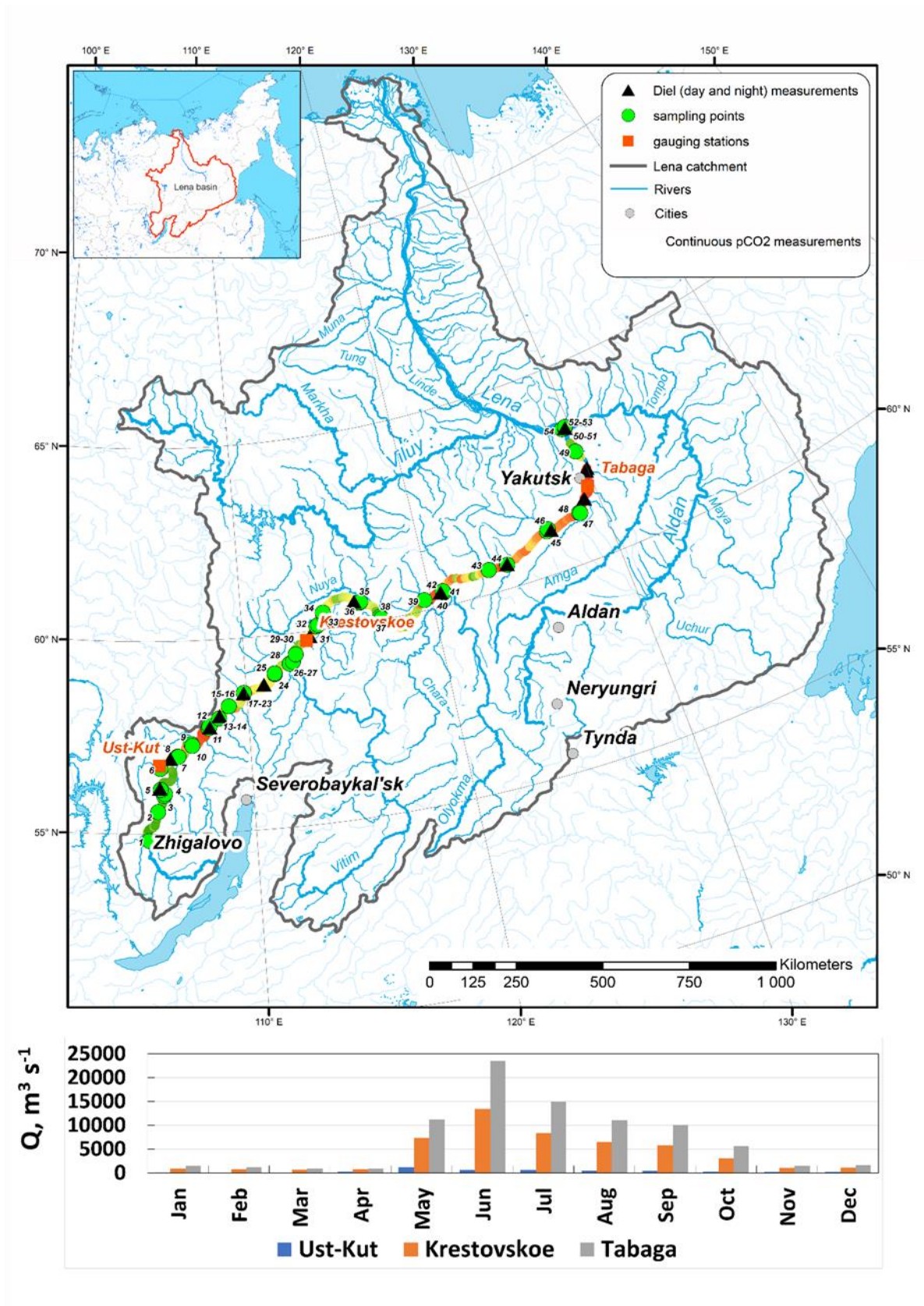


**Fig. 1.** Map of the studied Lena River watershed with continuous $pCO_2$ measurements in the main stem. Bottom: mean multi-annual monthly discharge (Q) at Ust-Kut, Krestovskoe and Tabaga station (labelled in red on the map).



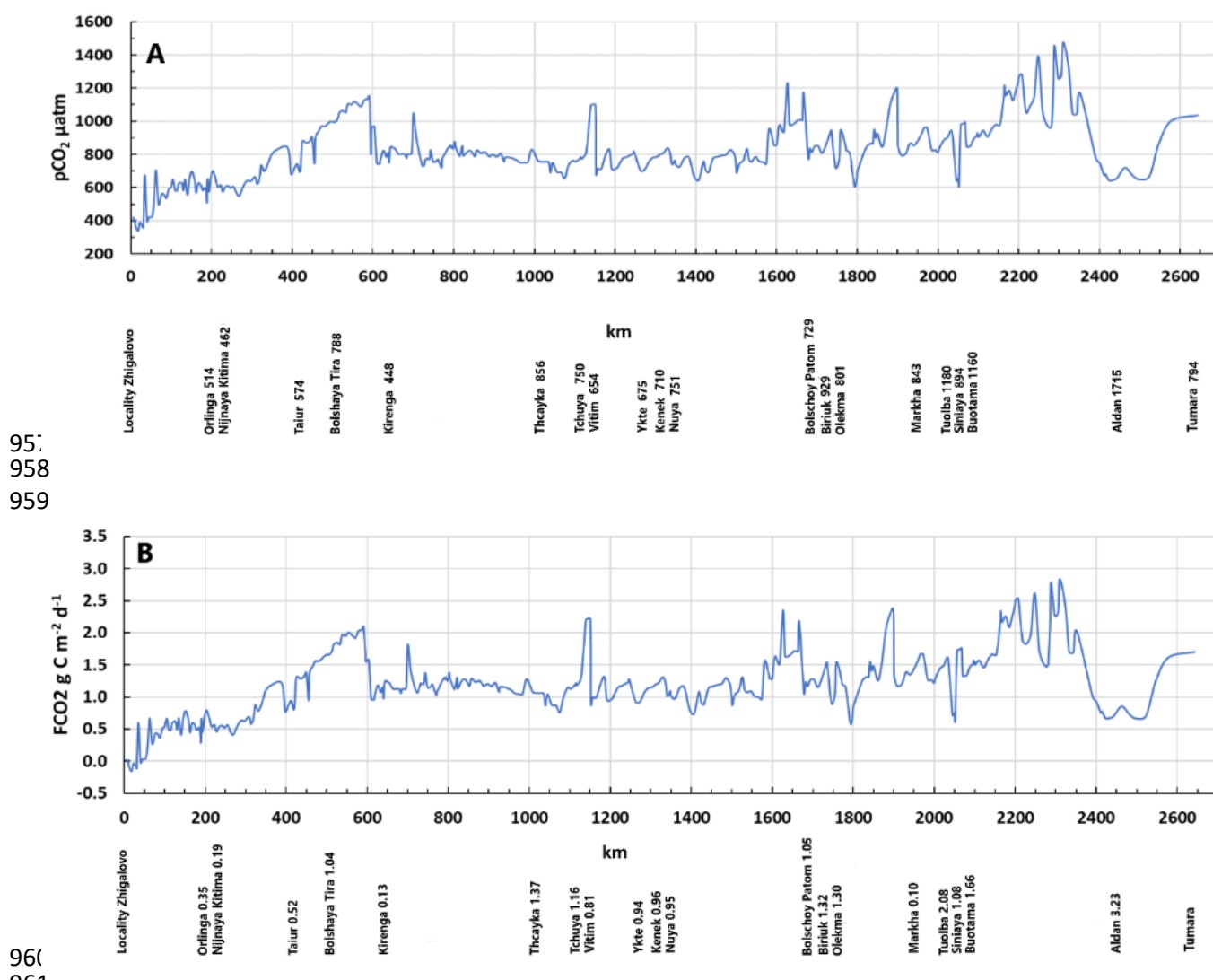

**Figure 2.** A 20-km averaged $pCO_2$ profile (**A**) and calculated $CO_2$ fluxes (**B**) of the Lena River main stem of over 2600 km distance, from Zhigalovo to the Tumara River. The average $pCO_2$ (µatm) and fluxes (g C m$^{-2}$ d$^{-1}$) of the main sampled tributaries are provided as numbers below X axes. Note that peaks of $CO_2$ concentration at the main stem are not linked to conflux with tributaries.

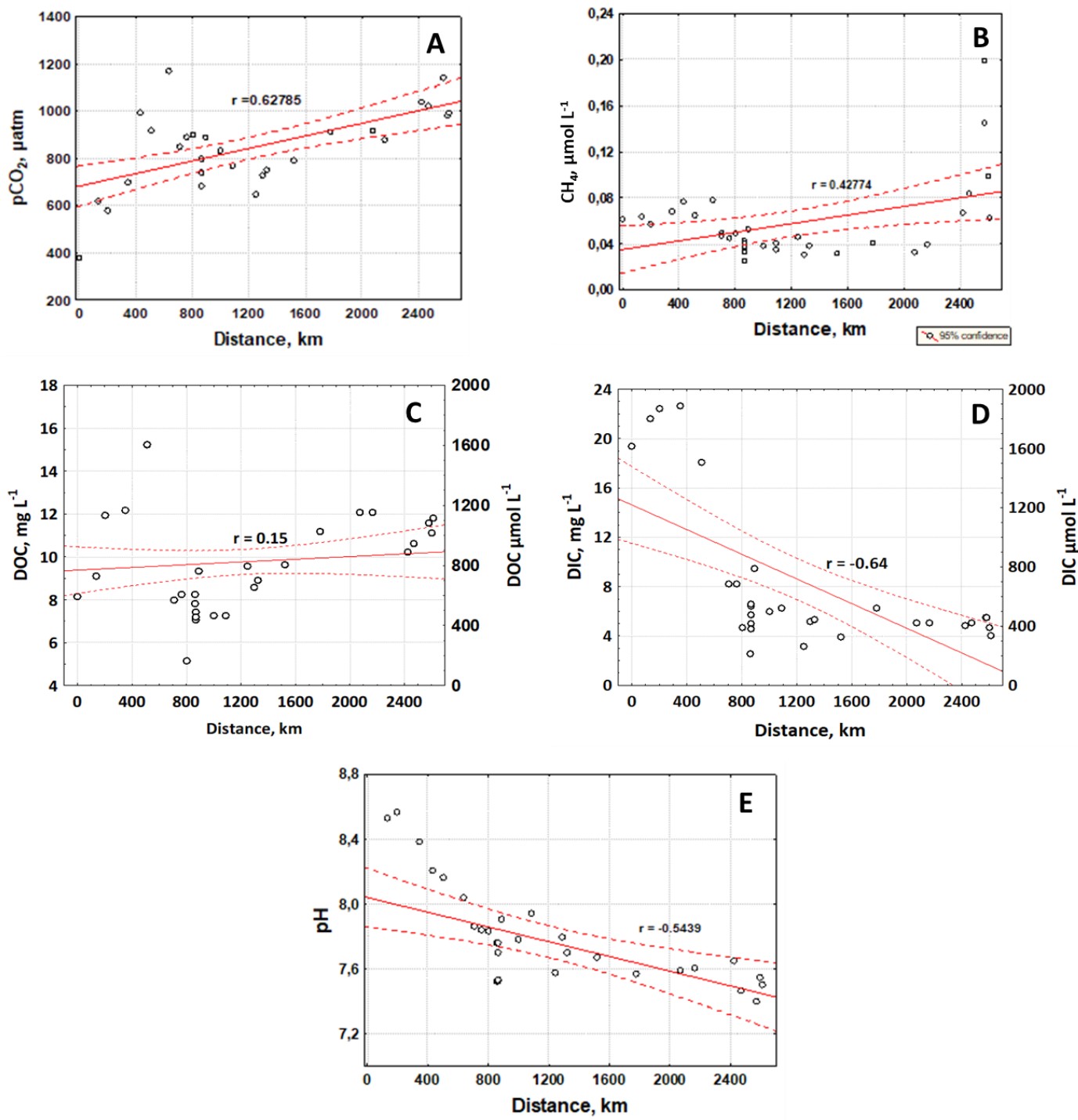


**Figure 3.** Averaged (over 20-km distance) $CO_2$ (A), $CH_4$ (B), DOC (C), DIC (D) and pH (E)

concentration over the distance of the boat route at the Lena River, from the south-west to north-east.


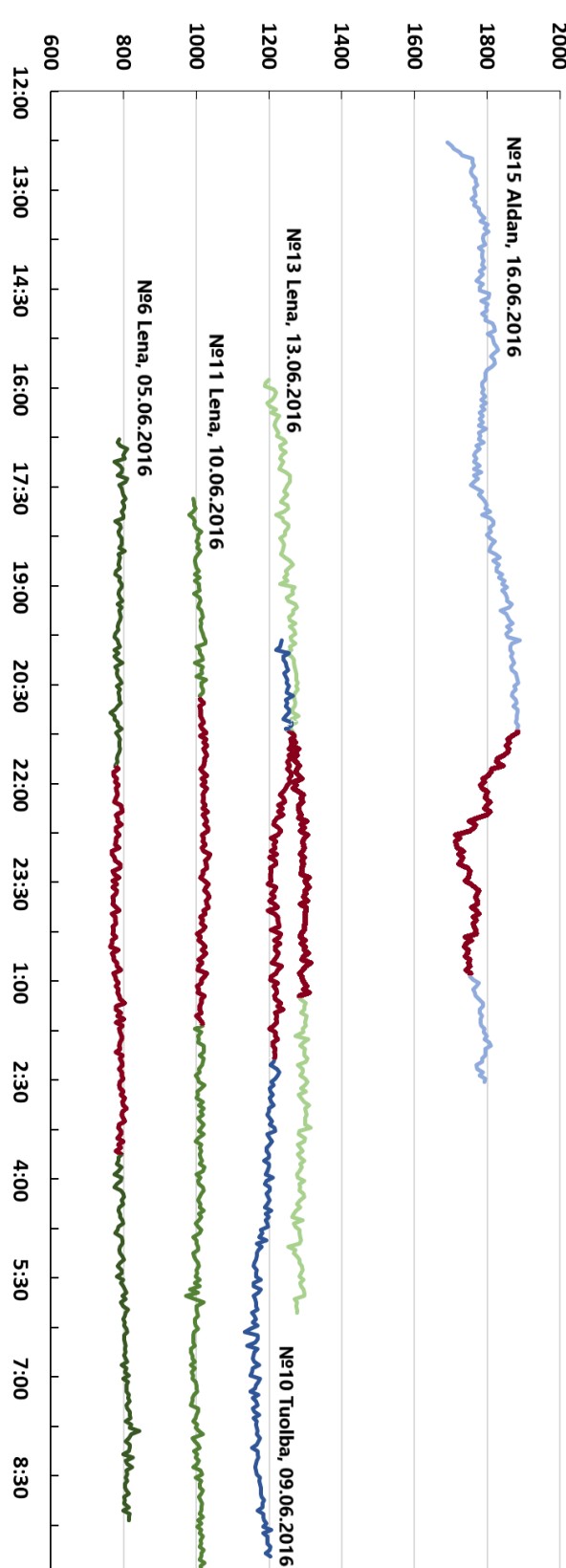

**Figure 4.** Continuous pCO$_2$ concentration in the Lena River and two tributaries from late afternoon to
morning next day. Red part of the line represents night time. Variations of water temperature did not
exceed 2 °C.

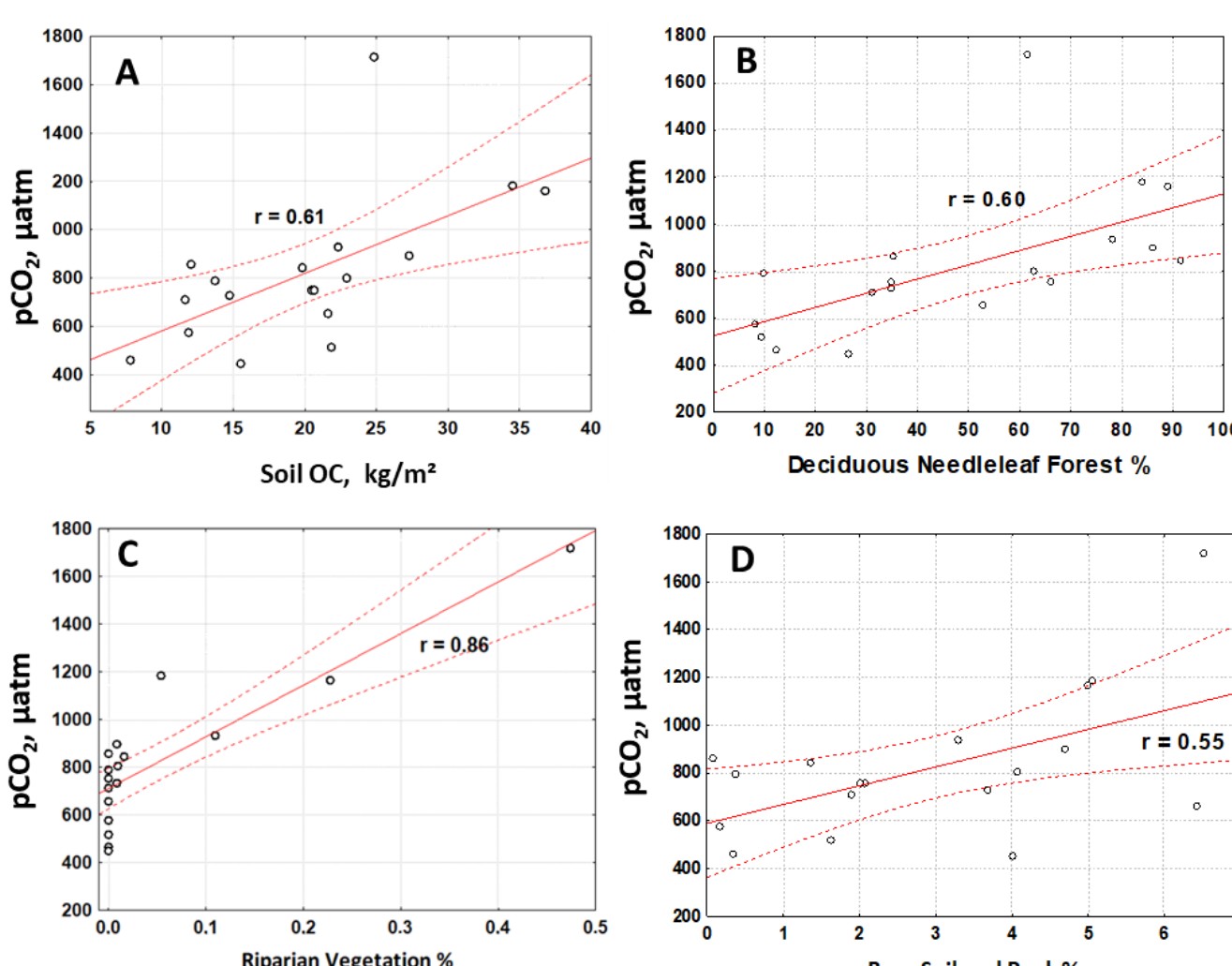

**Figure 5.** Significant ($p < 0.05$) positive control of landscape parameters – OC stock in 0-100 cm of soil (A), and proportion of deciduous needle-leaf forest (B), riparian vegetation (C) and bare soil and rock (D) in the watershed on $pCO_2$ in the Lena River tributaries.