# Peer review of "Fluvial carbon dioxide emission from the Lena River basin during spring flood"

_Biogeosciences, 2021_

## Author Comment (AC2)

**Table R1.** Measured water temperature, $pCO_2$, calculated $CO_2$ flux, $CH_4$, DOC, and DIC concentrations and pH in the Lena River main stem (average $\pm$ s.d.; (n) is number of measurements). The $CO_2$ emission fluxes ($FCO_2$) are calculated for two values of transfer coefficient ($k$) of 4.464 m d$^{-1}$ (Karlsson et al., 2021) and 3.00 m d$^{-1}$ (lowerst range of world rivers in Raymond et al., 2013).

| River transect | $T_{water}$, °C | $pCO_2$, µatm | $FCO_2$, g C m$^{-2}$ d$^{-1}$ $k$ = 4.464 | $FCO_2$, g C m$^{-2}$ d$^{-1}$ $k$ = 3.00 |
|---|---|---|---|---|
| Lena upstream of Kirenga (0-578 km) | 12.65±0.22 (99) | 714±22 (99) | 0.849±0.061 (99) | 0.571±0.041 (99) |
| Lena Kirenga – Vitim (579-1132 km) | 9.17±0.15 (87) | 806±8.8 (87) | 1.19±0.024 (87) | 0.802±0.016 (87) |
| Lena Vitim -Nuya (1132-1331 km) | 8.10±0.115 (27) | 797±22 (27) | 1.22±0.072 (27) | 0.817±0.048 (27) |
| Lena Nuya – Tuolba (1331-2008 km) | 9.61±0.09 (95) | 846±12 (95) | 1.29±0.034 (95) | 0.868±0.023 (95) |
| Lena Tuolba – Aldan (2008-2381 km) | 10.6±0.21 (52) | 1003±28 (52) | 1.69±0.081 (5) | 1.21±0.048 (52) |

| | $CH_4$, µmol L$^{-1}$ | DOC, mg L$^{-1}$ | DIC, mg L$^{-1}$ | pH |
|---|---|---|---|---|
| Lena upstream of Kirenga (0-578 km) | 0.068±0.003 (6) | 13.9±1.4 (6) | 20.0±1.2 (6) | 8.12±0.203 (7) |
| Lena Kirenga – Vitim (579-1132 km) | 0.040±0.002 (12) | 7.55±0.246 (14) | 6.30±0.485 (14) | 7.77±0.040 (14) |
| Lena Vitim -Nuya (1132-1331 km) | 0.038±0.003 (5) | 9.02±0.29 (3) | 4.55±0.70 (3) | 7.69±0.063 (3) |
| Lena Nuya – Tuolba (1331-2008 km) | 0.037±0.002 (6) | 10.4±0.78 (2) | 5.09±1.157 (2) | 7.62±0.052 (2) |
| Lena Tuolba – Aldan (2008-2381 km) | 0.088±0.034 (5) | 11.6±0.27 (5) | 5.24±0.102 (5) | 7.49±0.044 (5) |

**Table R2.** Measured water temperature, $pCO_2$, calculated $CO_2$ flux, $CH_4$, DOC, DIC concentration and pH in the tributaries (average ± s.d.; (n) is number of measurements). The $CO_2$ emission fluxes ($FCO_2$) are calculated for two values of transfer coefficient ($k$) of 4.464 m d$^{-1}$ (Karlsson et al., 2021) and 3.00 m d$^{-1}$ (lowest range of world rivers in Raymond et al., 2013).

| Tributary | $T_{water}$, °C | $pCO_2$, µatm | $FCO_2$, g C m$^{-2}$ d$^{-1}$ $k$ = 4.464 | $FCO_2$, g C m$^{-2}$ d$^{-1}$ $k$ = 3.00 |
|---|---|---|---|---|
| №4 Orlinga (208 km) | 8.0±0.0 (13) | 515±2.9 (13) | 0.347±0.01 (13) | 0.233±0.005 (13) |
| №5 Nijnaya Kitima (228 km) | 6.8±0.0 (11) | 462±9.4 (11) | 0.193±0.03 (11) | 0.130±0.020 (11) |
| №8 Taiur (416 km) | 8.5±0.0 (10) | 575±31 (10) | 0.523±0.095 (10) | 0.351±0.064 (10) |
| №10 Bol. Tira (529 km) | 11.9±0.0 (15) | 788±12 (15) | 1.04±0.03 (15) | 0.701±0.021 (15) |
| №12 Kirenga (579 km) | 10.2±0.0 (323) | 448±4 (323) | 0.131±0.01 (323) | 0.088±0.008 (323) |
| №25 Thcayka (1025 km) | 8.6±0.01 (8) | 856±13 (8) | 1.37±0.04 (8) | 0.922±0.026 (8) |
| №28 Tchuya (1110 km) | 5.9±0.0 (5) | 751±5.7 (5) | 1.16±0.019 (5) | 0.779±0.013 (5) |
| №29 Vitim (1132 km) | 6.8±0.0 (10) | 654±10 (10) | 0.812±0.03 (10) | 0.602±0.018 (10) |
| №32 Ykte (1265 km) | 4.9±0.0 (11) | 676±4.8 (11) | 0.943±0.02 (11) | 0.634±0.011 (11) |
| №34 Kenek (1312 km) | 7.60±0.0 (11) | 710±2.6 (11) | 0.964±0.01 (11) | 0.648±0.005 (11) |
| №36 Nuya (1331 km) | 11.8±0.0 (10) | 752±6.0 (10) | 0.947±0.02 (10) | 0.637±0.011 (10) |
| №38 Bol. Patom (1670 km) | 6.9±0.0 (5) | 730±12 (5) | 1.05±0.04 (5) | 0.706±0.026 (5) |
| №39 Biriuk (1712 km) | 14.2±0.0 (5) | 929±19 (5) | 1.32±0.05 (5) | 0.888±0.032 (5) |
| №40 Olekma (1750 km) | 6.4±0.0 (11) | 802±14 (11) | 1.30±0.05 (11) | 0.876±0.032 (11) |
| №43 Markha (1948 km) | 17.5±0.0 (15) | 844±15 (15) | 0.998±0.03 (15) | 0.671±0.023 (15) |
| №44 Tuolba (2008 km) | 12.3±0.0 (305) | 1181±6 (305) | 2.08±0.02 (305) | 1.395±0.010 (305) |
| №46 Siniaya (2118 km) | 18.5±0.0 (24) | 894±19 (24) | 1.08±0.04 (24) | 0.727±0.029 (24) |
| №48 Buotama (2170 km) | 18.5±0.0 (24) | 1160±25 (24) | 1.66±0.06 (24) | 1.118±0.037 (24) |
| №52-54 Aldan (2381 km) | 14.8±0.02 (316) | 1715±12 (316) | 3.23±0.03 (316) | 2.172±0.020 (316) |

**Fig. R1.** Pearson correlations between $pCO_2$ and DIC (A) and pH (B of the Lena River main stem (solid circles, solid line) and Lena tributaries (open circles, dotted line). Significant correlations ($p < 0.05$) are marked by asterisk.

---

## Author Response (AR1)

**Response to Editor**

- Lines 30-31: Not pCO2 but CO2 is oversaturated ("supersaturated"), so please rephrase the sentence, like "…exhibited CO2 supersaturation…."
**We totally agree and corrected the text accordingly.**

- pCO2 sensor measurements: Given the critical importance of sensor measurement accuracy, you need to provide more methodological details, including the used membrane, probe maintenance (cleaning surface fouling), boat speed, sensor accuracy validation, and data processing. From my own experience (refer to Fig. 2 of Park et al. Biogeochemistry https://doi.org/10.1007/s10533-021-00823-6 to see an example of boat speed effects on sensor pCO2 measurements) and other studies (Crawford et al. ES&T dx.doi.org/10.1021/es504773x; Yoon et al. Biogeosciences 13, 3915–3930), fast boat speed exceeding 10 km/hr can create too much turbulence for accurate sensor measurements.
**We agree with these pertinent remarks and we provided the following information in the methods: The key to aqueous deployment of the IRGA sensor is the use of a protective expanded polytetrafluoroethylene (PTFE) tube or sleeve that is highly permeable to $CO_2$ but impermeable to water (Johnson et al., 2009). The material is available for purchase as a flexible tube that fits over the IRGA sensor (Product number 200-07; International Polymer Engineering, Tempe, Arizona, USA). During the sampling, the sensor was left to equilibrate in the water for 10 minutes before measurements were recorded.**
**Note that sensor accuracy validation is described in the section Methods: "The measurement unit (MI70, Vaisala®; accuracy ± 0.2%) was connected to the sensor allowing instantaneous readings of $p$CO$_2$. The sensors were calibrated in the lab against standard gas mixtures (0, 800, 3 000, 8 000 ppm; linear regression with $R^2$ > 0.99) before and after the field campaign. The sensors' drift was 0.03-0.06% per day and the overall error was 4-8% (relative standard deviation, RSD). Following calibration, post-measurement correction of the sensor output induced by changes in water temperature and barometric pressure was done by applying empirically derived coefficients following Johnson et al. (2009). These corrections never exceeded 5% of the measured values."**

**Yoon et al. (2016) demonstrated that the membrane-enclosed sensor could be vulnerable to biofouling in polluted waters on the example of some anthropogenicallly-affected river of subtropical climate. These authors recommended to use a coppermesh screen to minimize the biofouling effects. In our sensor design, we also used the coppermesh screen be used to minimize the biofouling effects, and we added this information in the revised text. More importantly, the Lena river waters during spring flood are cold and virtually pristine and thus unlikely to develop any biofouling. Furthermore, we tested two different sensors in several sites of the river transect and never found any sizable (>10% difference) in measured $CO_2$ concentration. One probe was used as a control and never employed for continuous measurements. We did not find any sizable (>10% difference) in measured $CO_2$ concentration between two probes. Upon return to the laboratory, the probes were calibrated and the response was within 95% of the original calibration, which was explicitly taken into account when converting $CO_2$ concentrations into pCO$_2$.**

We have also tested the impact of the boat speed (5, 10, 20, 30 and 40 km h$^{-1}$) on the sensor performance and have not detected any sizable (> 10%) difference in the concentrations recorded by our system. We added this important information in the revised text (section 2.2, L 142-148).

Recently, on the Ob River, we used an alternative chamber design to test the impact of turbulence on the performance of our sensor (Fig. R1) during on-board measurements. We did not find sizable (> 20%) difference between "direct in the tube" and "no-buble low-turbulence chamber" measurements

1 plastic outboard inlet pipe
2 inlet silicone hose
3 electric pump 12 volt
4 water vessel
5 CO2 sensor with waterproof membrane
6 outlet gravity branch pipe
7 funnel with outboard outlet

**Fig R1.** An alternative chamber design allowing to maximally avoid the turbulence during in-situ CO$_2$ measurements by IRGA sensor during the boat movement.

I wondered if you could use some pCO$_2$ data from the manual headspace equilibration used for CH$_4$ measurements to validate your sensor measurements.
**This is quite a pertinent comment. The validation of CO$_2$ measurements by CARBOCAP probe of Vaisala in both fluid and gaseous environment was described in previous works of our group in the Western Siberia Lowland (Serikova et al., 2018, 2019) and in the seminal paper of Johnson et al. (2009). In our previous studies in the WSL lakes, we verified that the CO$_2$ analysis of headspace provided reasonable agreement with in-situ measuremenets by Vaissala. In the present study, the samples collected for CH$_4$ analysis were found to be unsuitable for CO$_2$ measurements; presumbaly, due to addition of high HgCl$_2$ concentration that could strongly affect the carbonate system equilibrium in carbonate-rich river waters (r$^2$ for the dependence of CO$_2$ concentration measured in samples of the Lena River by gas chromatography and by CARBOCAP probe of Vaisala was equal to 0.76, p < 0.01).**

You showed 20-km averaged pCO₂ profile in Fig. 2. Please describe how you processed original pCO₂ data (and based on what criteria?).

**The 20-km interval of the boat route represented an average of 3 consequitive slos of 5-min measurements and 10 min stand by of the probe; added to the revised text.**

**A comparison of direct pCO₂ data collected by the sensor (every minute during 5 minutes at 10 minutes interval) and 20-km averaged for a selected river transect (~400 km) is presented in Fig. R2.**

[Figure]

**Distance of the boat route, km from headwaters**

**Fig. R2.** Selected plot of primary pCO₂ data of the Lena River main stem (blue line) and approx. 20-km averaged 3 slots of 5 min with 10 min interval (orange line). Two slots of the boat route (200-300 and 600-1000 km from headwaters) are shown.

- Selection of k values: In your response to a reviewer comment on k, you indicated "Decreasing the k to even more conservative value of 3 m d-1". If you have to resort to literature values to provide a reliable range of k for your system, I would suggest that you provide the most representative range for northern rivers. For example, Lauerwald et al. (Global Biogeochem Cycles, 29, 534–554) used two ranges of k: streams and small rivers (4-5.26) and large rivers (2.25-2.63) of Boreal-Arctic Zone. If stream orders vary a lot, you can probably use different ranges from representative literature values, separately for the Lena main stem and tributaries.

**We agree that such an approach would be more rigorous than postulating one single *k* value for the whole Lena River basin. However, several lines of arguments support our choice of a single value for gas transfer coefficient. First, given that our study is a first - order approach incorporating direct measurements of pCO₂ in virtually unknown riverine system (see Fig. 1 in Lauerwald et al., 2015), and for consistency with previous measurements in both large and small rivers of the permafrost zone (Karlsson et al., 2021), we believe that the chosen value of 4.34 m d⁻¹ adequately reflects the gas tranfert**

for the Lena River basin. **Second, we would like to note that the value used in the present study is fully consistent with the range recommended by Lauerwald et al. (2015) for global large rivers and boreal rivers. For example, for the Yukon River system, most similar to that of upper and middle reaches of the Lena River, Lauerweld et al. (2015) recommended the range of $k$ between 3.7 and 4.2 m $d^{-1}$ whereas Striegl et al (2012) provided k = 3.1 m $d^{-1}$ for the main stem of Yukon and 5.2 m $d^{-1}$ for the tributaries. Overall, we believe that decreasing the $k$ value of large Boreal-Arctic rivers to 2.25-2.63 m $d^{-1}$ (Lauerweld et al., 2015) which is sizably lower than that recommended by the same author for global large rivers (3.4-4.41) and measured by our group in the Ob River basin is especially unwarranted in the case of the Lena River because the latter exhibits sizable turbulence of the main stem and tributaries.**
*Lauerwald, R., G. G. Laruelle, J. Hartmann, P. Ciais, and P. A. G. Regnier (2015), Spatial patterns in $CO_2$ evasion from the global river network. Global Biogeochem. Cycles, 29,534–554, doi:10.1002/2014GB004941.*

- Chemical analyses other than $CO_2$ and CH4: Please provide essential information about the conducted in situ water quality measurements (pH,,,) and laboratory chemical analyses (DOC, DIC,,,), including the used instruments and QA/QC.

**Thank you for pointing this out. We added extensive description of relevant analyses in the Methods section (new section 2.3)**

- Your response regarding the pH-pCO2 relationship (The pH did not control the CO2 concentration in the main stem, and only weakly impacted the CO2 in the tributaries (Fig. R1 B).): The poor correlation may have resulted from various sources, including inaccurate measurements of pH (Limnol. Oceanogr.: Methods 18, 606–622) and pCO2 (refer to my previous comment) and the well-known effect of organic acids (Abril et al. Biogeosciences, 12, 67–78). Please discuss potential sources of measurement error together with your explanations.

**We believe that poor correlation reflects multiple sources of dissolved $CO_2$ in the river water such as *i*) respiration of phytoplankton and periphyton; *ii*) lateral influx of $CO_2$-rich soil waters and suprapermafrost waters; *iii*) heterotrophic degradation of riverine dissolved and particulate organic matter. The superposition of multiples sources, especially pronounced at high water level (flooding of the riparian zone) and water velocity of the river water likely create these highly non-equilibrium conditions. The pCO$_2$ and pH are expected to be correlated in $CO_2$-equilibrated waters, which is, presumbaly, not the case of the Lena river main stem and tributaries. Note that unlike in Abril et al work, here we measured the DIC via Simadzu method, not by Alkalinity titration. The organic ligands can indeed produce sizable artifacts of DIC measurements via Alkalinity titration, which was not the case of the present study. Concerning the uncertainties on pH measurements, they did not exceed 0.02 pH units. The measurements were fairly stable, given high $HCO_3$ concentration in the Lena River water which acted as a carbonate buffer.**

- The relative contribution of $CH_4$ to C emissions: Please provide some discussion and concluding remark on the relative significance of $CH_4$. It would also be helpful if you provide CO2eq estimates based on the global warming potential of $CH_4$.

**This is very interesting propostion. As we stated in the end of section 3.1, "The river water concentrations of dissolved $CH_4$ in the tributaries and the main channel (0.059±0.006; IQR range from 0.025 to 0.199 µmol $L^{-1}$, Table 1, 2) did not exhibit any trend with the distance from headwaters or the landscape parameters of the catchments. For these reasons, we are reluctant to discuss the cause of lack of landscape and geographical control on dissolved methane pattern in the Lena River basin.**

**These values are consistent with the range of $CH_4$ concentration in the low reaches of the Lena River main channel (0.03-0.085 µmol $L^{-1}$; Bussman, 2013) and 100-500 times lower than those of $CO_2$. Consequently, diffuse $CH_4$ emissions constituted less than 1 % of total C emissions and are not discussed in further details. Following the suggestion of the editor, we added the following text in the Discussion:**

**Typical concentrations of $CH_4$ in the Lena tributaries and the main channel are 100 to 500 times lower than those of $CO_2$. Given that the global warming potential (GWP) of methane on a 100-year scale is only 25 times higher than that of $CO_2$, the long-term role of diffuse methane emission from the Lena River basin is still 4 to 20 times lower than that of $CO_2$. However, on a short-term scale (20 years), the GWP of methane can be as high as 96 (Alvarez et al., 2018) and its role in climate regulation becomes comparable to that of the $CO_2$. This has to be taken into account for climate modeling of the region.**

*Alvarez et al (2018). "Assessment of methane emissions from the U.S. oil and gas supply chain". Science. 361 (6398): 186–188. doi:10.1126/science.aar7204*

**Reviewer No 1**

REVIEWER: The reviewer No 1 correctly pointed out that "some of the conclusions draw by the article to be lacking complete discussions and support by references and other supporting ideas".

RESPONSE: In fact, the present study was not designed to address the mechanisms of $CO_2$ generation in the Lena River main stem and tributaries. Such an investigation would require quite different sampling and measurement design. We would like to note that some discussion on $CO_2$ -related processes is provided in L 302-336, whereas thorough comparison with relevant literature data is given in section 4.2. In response to this comment, we added two plots of $CO_2$-DIC and $CO_2$-pH dependences for the Lena River. Furthermore, all the relevant landscape parameters are listed in Table 3 and discussed in section 4.1. We also extended the discussion of PCA results in the section 4.1

In the revised version, we extended the discussion and provided necessary references.

REVIEWER: Some of the parameters discussed in the methods and results section are not discussed in the discussion section. Discussion of these parameters would strengthen the arguments made by the authors.

RESPONSE: The reviewer made a good point here. However, most of these parameters turned out to be non-correlated to $pCO_2$ in the river water. As such, there is no reason to discuss the lack of control by this or that environmental parameter given that we cannot ascertain the reason for this case. All the relevant landscape parameters are listed in Table 3.

REVIEWER: PCA results are presented in the results sections with no description in the methods section. The PCA results should be revisited in the discussion section.

RESPONSE: We totally agree with this pertinent comment and we added necessary methodological description in the end of section 2.5. We would also like to provide more discussion on the PCA results. However, the PCA demonstrated extremely low ability to describe the data variability (12% by F1 and only 3.5% by F2). We believe that the most likely reason of weak PCA capacity is rather homogeneous distribution of $CO_2$ and $CH_4$ across the river transect and among tributaries, primarily linked to the specific hydrological period, studied in this work - the springflood. During this high flow period, the local lithological and soil heterogeneities among tributaries or the segments of the main stem virtually disappear and the surface flow (via vegetation leaching) becomes most important driver of riverine chemistry, as it is known from adjacent permafrost territories of Central Siberia (i.e., Bagard et al., 2011). Nevertheless, some specific features of the data structure could be established. The first factor, significantly linked to $pCO_2$ (0.72 loading), strongly acted on the sample location at the Lena transect, the watershed coverage by deciduous needle-leaf forest and shrubs, riparian vegetation, but also the proportion of tundra, bare rock and soils, water bodies, peatland and bogs (> 0.90 loading). This is fully consistent with spatial variation of $pCO_2$ along the permafrost and climate gradient in the main channel and sampled tributaries. Positive loading of riparian vegetation, peatlands and bogs on F1 (0.927 and 0.989, respectively) could reflect a progressive increase in the feeding of the river basin by mire waters, increase in the proportion of needle-leaf deciduous trees, and an increase in the width of the riparian zone from the SW to the NE direction. We added necessary discussion in the revised section 4.1.

Methods: For the PCA treatment, all the variables were normalized as necessary in standard package of STATISTICA-7 (http://www.statsoft.com) because the units of measurements of various components were different. The factors were identified via the Raw Data method. To run the scree test, we plotted the eigenvalues in descending order of their magnitude against

their factor numbers. There was significant decrease in the PCA values between F1 and F2 suggesting therefore that maximum two factors were interpretable.

REVIEWER: The reviewer also stated that 'On line 322, the authors suggest that in-stream processing of dissolved terrestrial organic C is not the main driver of $CO_2$ supersaturation in the river waters of the Lena River basin, but offer no alternative pathways for this phenomenon.'
RESPONSE: The relevant mechanisms of $CO_2$ supersaturation are discussed in L327-334. We extended this discussion in the revised version as following. The main sources of $CO_2$ in the river water include but not limited to *i*) hyporheic discharge of $CO_2$-rich underground waters, *ii*) lateral influx of $CO_2$-rich soil waters; *iii*) dissolved and particulate organic carbon processing in the water column via bio- and photodegradation, and iv) phyto, zoo-plankton, periphyton and sediment respiration. As indicated in the text (L327-331), there was no relationship ($p < 0.05$) between the proportion of carbonate rocks on the watershed and the $pCO_2$ in the tributaries (**Fig. S6 B**), whereas for the Lena River main stem, the lowest $CO_2$ concentrations were recorded in the upper reaches (first 0-800 km) where the carbonate rocks dominate the background lithology. This makes unlikely the impact of underground $CO_2$ from carbonate reservoirs on river water $CO_2$ concentrations. Given that we have not recorded any sizable diurnal variations in $pCO_2$ over the full transect of the Lena River, the respiration of photosynthetic organisms (plankton and periphyton) cannot be the reason for persistent $CO_2$ supersaturation over day and night. Furthermore, there was no significant ($p < 0.05$) link between DOC and $CO_2$ concentration, so we do not expect sizable impact of bio- and photodegradation of DOM. A lack of lateral (across the river bed) variations in $pCO_2$ witnesses against sizable input of soil waters from the shore, although we admit that much higher spatial coverage along the river shore is needed to confirm this hypothesis. Therefore, other sources of riverine $CO_2$ may include particulate organic carbon processing in the water column (Attermeyer et al., 2018), river sediments (Humborg et al., 2010) and within the riparian zone (Leith et al., 2014, 2015). Quantifying these impacts at the scale of the Lena River basin will certainly require further investigation.

REVIEWER: The reviewer also noted that the text 'needs to be reviewed and edited by a native English speaker'. The revised text was subjected to thorough editing by a native English speaking scientist. We would like to point out that the APC of Biogeosciences include thorough English style and grammar revision, and we hope to use this option for our manuscript.

Specific comments of Reviewer No 1:

Line 331 POC is not defined. Response: Particulate Organic Carbon, was added to revised text.

Line 344 FCo2 not defined. Response: This is $CO_2$ emission flux; corrected.

Line 344 Unites should be United. Response: We are sorry for this misprint and corrected it accordingly.

**Reviewer No 2**

REVIEWER: The reviewer No 2 correctly argued that 'some of the drawn conclusions are lacking proof, and likely overestimate the annual carbon emissions.'
RESPONSE: We revised our conclusions and estimations, following his/her detailed comments below.

REVIEWER: Comment to line 167-169. A fixed $k_{CO_2}$ value for the entire open water season of 4.6 m day $^{-1}$ is rather high, especially since floating chambers often overestimate the fluxes (Long et al., 2017; Ribas-Ribas et al., 2018). Particularly when using floating chambers during the freshet, where the water velocity and turbulences are several times above the summer low which then lasts for 4 to 5 months. Used reference measured a median of 4,464 m d$^{-1}$, which were all sampled during June. In addition, since many $k$ measurements were made, I would suggest separating main stems and tributaries. Also, when looking up the $k$ values from the given ref. Serikova et al., all reported $k$ values were given in cm$^{-1}$ h$^{-1}$, ranging between 5.1 and 16.5 cm$^{-1}$ h$^{-1}$ (which is 1.2 to 4 m day$^{-1}$). Please double check that the proper k value unit was used.
RESPONSE: This is very pertinent comment. In our calculations, we used a fixed value of 4.464 m d$^{-1}$ which represents the average of Ob, Pur and Taz Rivers by Karlsson et al. (2021). These rivers are similar to Lena and its tributaries in size, but exhibit lower velocity and turbulence than those of the Lena River basin. In fact, due to more mountainous relief, the Lena River main stem and tributaries have much higher turbulence than that of the Ob River and tributaries and as such this estimation can be considered rather conservative. Decreasing the $k$ to even more conservative value of 3 m d$^{-1}$ (which is the lowest range of world's rivers as recommended by Raymond et al., 2013) provide the values of specific emissions which are 30 to 50% lower than those obtained in this study ($k$ = 4.464 m d$^{-1}$). The resulted corrections in aerial emissions yield the from value ranging between 0.8 and 1.5 g C m$^{-2}$ d$^{-1}$ corresponding to total value of 4 to 7.5 Tg C y$^{-1}$. For convenience, we attached the revised tables to this response (Tables R1 and R2). Note that main stem and tributaries are always separated in the text, figures and tables (see Tables R1 and R2 below). Please see our response to Editor's comment on $kCO_2$ values and the possibility of using the data of Lauerwald et al. (2015). Note that the $kCO_2$ values in Serikova et al. (2018) are relevant to small rivers of the western Siberia. Such small rivers constitute less than 20% of the water surfaces in the Lena basin (see section 4.2 of our revised manuscript). As such, we preferred using the k values obtained on large (Ob) and medium (Pur, Eaz) size rivers of the permafrost region, which are most similar to the Lena main stem and its large tributaries, studied in this work.

REVIEWER: Comment to section 3.4 on aerial emissions. As your own data shows, there are strong temporal and spatial variability in $pCO_2$ levels.
RESPONSE: We do not completely agree with this statement. As we show in our work, the $pCO_2$ in Lena and tributaries remain generally stable over the night and day period (Abstract, Fig. 4, Fig. S2). The lateral variability over the tributaries and across the channel is also low (Fig. S1B, Fig. S3). The global variability in $pCO_2$ over the largest part (~2400 km) of the main stem is "only" ±20% (from 800 to 1200 µatm, see Fig. 2 A). The variability of $pCO_2$ in the tributaries is indeed, higher (from 600 to 1100 µatm) and this explicitly taken into account during our overall estimations of C emissions (see revised section 3.4).

REVIEWER: Upscaling spring flood concentrations, where >50% of annual water masses discharges, for the remaining 4 summer months is highly uncertain. Summer concentrations from e.g. the Kolyma are reported to be 0.35 g C m$^{-2}$ d$^{-1}$. Also, in line 266 you report that

5022 km$^2$ water area are seasonal. This area needs to be removed when calculating the areal summer fluxes.

RESPONSE: We agree with sizable uncertainty on our estimations, which amounts to ca. 50% (from 1 to 2 g C m$^{-2}$ d$^{-1}$). We demonstrate, via analysis of available literature data, that seasonal variations of pCO$_2$ in the Lena River main stem do not exceed the range of our uncertainties (section 3.4, L 270-279). We do acknowledge sizable uncertainties on our first order estimations, especially in view of lack of direct pCO$_2$ data for the northern tributaries including a very large river Vilyi (L377-382). We further agree that rigorous aerial estimation should include 4 summer months with lower surface water coverage. However, introducing this correction changes the global value by only 12% which is below the range of our uncertainties; see the last paragraph of section 3.4.

REVIEWER: Comment to line 358ff: What published data and I would like to see a table with this literature data. What are the numbers? If available with seasonal resolution as this is what you are comparing with.

RESPONSE: Extensive description of all the relevant literature data is provided in section 3.4, L 270-279 (2$^{nd}$ paragraph of section 3.4 in the revised version). We believe that adding an explicit table will lengthen the paper and preferred to use the current format which is easier for the reader.

REVIEWER: Comment to the discussion section. Especially here English needs to be revised and restructured. Some parts can be shortened, while several other parameters which were introduced, were not discussed at all.

RESPONSE: We agree and reorganized this sections and revised the English. The three parameters of the river water chemistry (pH, DOC and DIC) were indeed, only partially discussed in the manuscript (L320-321, Fig. S6A). The correlation of pCO$_2$ with DIC and pH was not pronounced (see new Fig. S7 of the revised manuscript). The pH did not control the CO$_2$ concentration in the main stem, and only weakly impacted the CO$_2$ in the tributaries (Fig. S7 B). Such a weak control could reflect an increase in pCO$_2$ in the northern tributaries which exhibited generally lower pH compared to the SW tributaries; the latter draining through carbonate rocks. Lack of correlations of CO$_2$ with DIC and pH was consistent with generally low predictive capacity to calculate pCO$_2$ from measured pH, temperature and alkalinity as stated in L 280-281: the ratio of calculated to measured pCO$_2$ was 0.67±0.15 (n = 47). This, again, demonstrates highly dynamic and non-equilibrium behavior of CO$_2$ in the river waters, with possible local hot spots from lateral input of CO$_2$-rich soil or suprapermafrost waters. For these reasons, in-situ, high spatial resolution measurements of CO$_2$ concentration in rivers such as those reported in this study of the Lena Basin, are crucially important for quantifying the C emission balance in lotic waters of high latitudes.

REVIEWER: Figure 1 and S1 A: Since you have graticules, you do not need a north arrow. Actually, your north is not always "up" on the figures. Please remote them.

RESPONSE: Agree and edited accordingly.

REVIEWER: Figure S1 A: Change Landscape to Landcover map. Also, reference for this data.

RESPONSE: Agree and edited accordingly. The land cover information sources are described in section 2.4 (L183-191) and we are presented now in the Figure caption of the revised version.

REVIEWER: Figure 2. This data is very interesting, but what I am missing is the discussion on that. Are the peaks where conflux occurs? Higher fluxes due to turbulences? More information on differences between the tributaries.

RESPONSE: This is a good point. We do not have straightforward explanation for peaks shown on the diagram of the main stem. These peaks are not necessarily linked to $CO_2$-rich tributaries but likely reflect local processes in the main stem, including lateral influx from the shores and shallow subsurface waters, typical for permafrost regions of forested Siberian watersheds (i.e., Bagard et al., 2011). Given that the data were averaged over 20-km distance, these peaks are not artifacts but reflect local heterogeneity of the main stem (turbulences, suprapermafrost water discharge, sediment resuspension and respiration. Note that such a heterogeneity was not observed in the tributaries, at least at the scale of our spatial coverage (see Fig. S2, S3).

Bagard, M. L.; Chabaux, F.; Pokrovsky, O. S.; Viers, J.; Prokushkin, A. S.; Stille, P.; Rihs, S.; Schmitt, A. D.; Dupre, B. Seasonal variability of element fluxes in two Central Siberian rivers draining high latitude permafrost dominated areas. Geochim. Cosmochim. Acta 75, 3335-3357, 2011.

Concerning the last remark of this question of the reviewer, the differences between tributaries (presentation of results and their discussion) make the central part of our study, and this information is provided in section 3.3 and 4.1.

REVIEWER: Table 1: $CH_4$ concentrations are illustrated twice. Please remove or exchange one.

RESPONSE: Thanks a lot for catching this! Corrected accordingly.

REVIEWER: Organic C and OC, choose one and use consistently.

RESPONSE: We homogenized as OC.

REVIEWER: Additional data from tables (DIC, pH) not really discussed and incorporated.

RESPONSE: The correlations of $pCO_2$ with DIC and pH were poorly pronounced (see response above) and as such neither DIC nor pH could serve as sole controlling factors of $CO_2$ concentration in the Lena River main stem and tributaries. However, following the recommendations of Editor and reviewers, we added new Fig. S7 together with relevant discussion (section 4.1) in the revised version.

**Reviewer No 3**

REVIEWER: This study presents a very interesting dataset. There is a significant lack of data on GHG emissions during the spring flood of Arctic rivers, so the data collected and presented is very insightful. Because of this I recommend putting in some extra work to make the most of the data, streamline this paper and make the conclusions stronger.
RESPONSE: We thank the reviewer for positive evaluation of our work and we revised the data presentation and interpretation as recommended.

REVIEWER: Comment to line 170: You change to comparing to $k_{600}$ values from literature, which is not the same as k values but you do not define $k_{600}$.
RESPONSE: We thank the reviewer for pointing out this inconsistency. In this study, we used the value of $k_t$ (a median gas transfer coefficient) of 4.464 m $d^{-1}$ measured in 4 largest rivers of Western Siberia in June 2015 (Ob', Pur, Pyakupur and Taz rivers; Karlsson et al., 2021). To standardize $k_t$ to a Schmidt number of 600, we used the following equation (Alin et al., 2011; Vachon et al., 2010):

$$k_{600} = k_t \, (600/Sc_{CO2})^{-n}$$

where $Sc_{CO2}$ is $CO_2$ Schmidt number for a given temperature ($t$, °C) in the freshwater (Wannikhof, 1992):

$$Sc_{CO2} = 1911.1 - 118.11t + 3.4527t^2 - 0.041320t^3$$
$$Sc_{CO_2} = 1911.1 - 118.11t + 3.4527t^2 - 0.041320t^3$$

and exponent $n$ is a coefficient that describes water surface (2/3 for a smooth water surface regime while 1/2 for a rippled and a turbulent one), and the Schmidt number for 20°C in freshwater is 600. We used $n = 2/3$ because all water surfaces of sampled rivers were considered flat and had a laminar flow (Alin et al., 2011; Jähne et al., 1987) and the wind speed was always below 3.7 m $s^{-1}$ (Guérin et al., 2007). We added necessary explanations in the text along with relevant references.

Alin, S. R. *et al.* Physical controls on carbon dioxide transfer velocity and flux in low-gradient river systems and implications for regional carbon budgets. *J. Geophys. Res.* **116,** G01009 (2011).
Guérin, F., Abril, G., Serça, D., Delon, C., Richard, S., Delmas, R., Tremblay, A., Varfalvy, L., 2007. Gas transfer velocities of CO2 and CH4 in a tropical reservoir and its river downstream. J. Mar. Syst., **66,** 161–172. https://doi.org/10.1016/j.jmarsys.2006.03.019
Jähne, B., Heinz, G. & Dietrich, W. Measurement of the diffusion coefficients of sparingly soluble gases in water. *J. Geophys. Res. Ocean.* **92,** 10767–10776 (1987).
Vachon, D., Prairie, Y. T. & Cole, J. J. The relationship between near-surface turbulence and gas transfer velocity in freshwater systems and its implications for floating chamber measurements of gas exchange. *Limnol. Oceanogr.* **55,** 1723–1732 (2010).

REVIEWER: Comment to line 176-177: Why did you use air concentrations from Mauna Loa Observatory and not closer stations such as Cherski or Barrow? What pCO₂ air concentration values were used to calculate the fluxes?
RESPONSE: The use of world medium $CO_2$ concentrations for gas transfer fluxes from water surfaces is the most standard approach in this field and we did so for consistency with numerous previous works. In this study we used pCO₂ = 402 ppm. It represents the average of 129 stations all over the world (World Meteorological Organization, 2009: Technical Report of Global Analysis Method for Major Greenhouse Gases by the World Data Center for Greenhouse Gases (Y. Tsutsumi, K. Mori, T. Hirahara, M. Ikegami and T.J.Conway). GAW

Report No. 184 (WMO/TD-No. 1473), Geneva, https://www.wmo.int/pages/prog/arep/gaw/documents/TD_1473_GAW184_web.pdf) taken from The World Data Centre for Greenhouse Gases (WDCGG) which is a World Data Centre (WDC) operated by the Japan Meteorological Agency (JMA) under the Global Atmosphere Watch (GAW) programme of the World Meteorological Organization (WMO). WDCGG (World Data Centre for Greenhouse Gases) (kishou.go.jp) https://gaw.kishou.go.jp Specifically, for the year of this study (2016) the world monthly average $CO_2$ concentration is as following (https://community.wmo.int/wmo-greenhouse-gas-bulletins):

| Year | Month | $pCO_2$ |
|------|-------|---------|
| 2016 | 1 | 403.34 |
| 2016 | 2 | 403.84 |
| 2016 | 3 | 404.35 |
| 2016 | 4 | 404.45 |
| 2016 | 5 | 404.16 |
| 2016 | 6 | 403.07 |
| 2016 | 7 | 401.51 |
| 2016 | 8 | 400.66 |
| 2016 | 9 | 401.39 |
| 2016 | 10 | 402.99 |
| 2016 | 11 | 404.43 |
| 2016 | 12 | 405.39 |

Thus, taking the period of this study, end of May - beginning of June, the average value is 402 ppm which was used in our calculations. This value is consistent with that directly measured at the Tiksi station in 2016: 404±0.9 ppm (Ivakhov et al., 2019). We added necessary explications in the section 2.4.

Ivakhov, V. M., Paramonova, N. N., Privalov, V. I., Zinchenko, A. V., Loskutova, M. A., Makshtas, A. P., Kustov, V. Y., Laurila, T., Aurela, M., and Asmi, E.: Atmospheric Concentration of Carbon Dioxide at Tiksi and Cape Baranov Stations in 2010–2017, Russian Meteorol. Hydrol., 44(4), 291–299, DOI: 10.3103/S1068373919040095, 2019.

REVIEWER: Comment to section 3.3: The discussion of the correlation of pCO2 with landscape parameters is not entirely consistent from the results to the conclusion. For example according to Table 3 pCO2 is correlated with riparian vegetation, but later on in the conclusion it is stated that it is correlated with the width of the riparian zone. So the riparian vegetation is a proxy for the width of the riparian zone? I note you did these correlations for the tributaries which gives interesting results, but how about for the main stem? It would be interesting to see since in the main stem pCO2 increases from south to north. The first sentence in the results section (L247-250) gives to understand that you did this but based on the captions of Table 3 and Figure 5 you only did the correlations with data from the tributaries- correct?
RESPONSE: The reviewer is totally correct. Yes, the riparian vegetation is a proxy for the width of the riparian zone; we added an explicatory sentence. And yes, we run the landscape control correlations only for the tributaries. The size and huge diversity of the main stem watershed did not allow producing sufficient information on land cover of the Lena River and this can be a subject of another study.

REVIEWER: Comments to section 3.4: The calculations of the areal lotic C emission for the entire open water season are not entirely clear to me. Did you use different pCO2 and k values for the main stem and the tributaries? You state that 1 to 2 g C m-1 d-1 covers full variability of the large and small tributaries and the Lena River main channel (L291-293) but Tables 1-2 show that there is values lower and higher than this. Also L348 states that the range in the tributaries is (0.2 to 3.2 g C m-2 d-1) and L289 that the Aldan river had considerable higher emissions than Lena river main stem, how was this taken into consideration in the areal C emission calculation?

RESPONSE: This is very pertinent comment and we thank the reviewer for bringing it out. The $k$ value for the main stem and tributaries was the same (4.46 m d$^{-1}$); it represents the average value measured in four largest rivers of Western Siberia in June 2015 (Ob', Pur, Pyakupur and Taz rivers, Karlsson et al., 2021). These four rivers are fairly representative for the Lena River and its tributaries, although the $k$ value should be considered as highly conservative (see our responses to Editor and reviewer No 2). The $pCO_2$ used for flux calculation (Table 2) was directly measured in the full transect of the main stem and the tributaries. When providing the largest possible span of average emission values (1 to 2 g C m$^{-2}$ d$^{-1}$), we used the median values of the main stem and tributaries.

We further revised the calculations following the comments of this and other reviewers. For this, we explicitly took into account the water area of the main stem (43%) relative to the total Lena basin and we introduced the partial weight of emission from three largest tributaries (Aldan, Olekma and Vitim) according to their catchment surface areas (43, 12 and 14% of all sampled territory, respectively). We summed up the contribution of the Lena river main stem and the tributaries and we postulated the average emission from the main stem upstream of Aldan (1.25±0.30 g C m$^{-2}$ d$^{-1}$) as representative for the whole Lena River. This resulted to updated value of 1.65±0.5 g C m$^{-2}$ d$^{-1}$ which is within the range of 1 to 2 g C m$^{-2}$ d$^{-1}$ assessed previously. Note that this value is most likely underestimated because the emissions from the main stem downstream of Aldan are at least 10 % higher (Table 1, Fig. 1 B), and it could be so for the whole remaining part of the basin, not sampled in this work.

REVIEWER: In terms of the k values used: You answered to the comment from reviewer 2 that you use the k values 4.46 m/d from Karlsson et al., 2021, this is not clear in L167-169. It reads as if you used the value 4.6 m/d based on Serikova et al., 2018 and Karlsson et al., 2021. You do then in L218 state that 4.46 m/d from Karlsson et al., 2021 is used. I would suggest changing L167-169 so this is consistent.

RESPONSE: Good point; thanks for catching this. We have corrected all numbers and revised the text accordingly. One single value of k (4.464) was used throughout all calculations.

REVIEWER: The dataset collected is very interesting and provides a lot of great insights. To me it is a bit of a missed opportunity to not utilise it more when estimating the areal lotic C emissions of the Lena basin. I would have liked to see how estimated areal CO2 emissions during the spring flood months, calculated with a k value corresponding to the higher flow, a larger water surface area (281000 km2) and your slightly higher pCO2 values compare to the summer month, calculated with a k value corresponding the lower flow, a smaller water surface area (281000 km2 - 5022 km2) and previously published slightly lower pCO2 values. I note that you replied to reviewer 2 that decreasing the water surface area for summer reduced the result by less than 15% which is below the range of your uncertainty. It would be good to see this more explicitly in the publication, this is not clear in section 3.4. Is this what the number 0.67±0.15 (n = 47) (L281) indicates? It would also be interesting to know how

much the Lena River main stem contributes to the areal CO2 emissions in contrast to the tributaries.

RESPONSE: We agree with a necessity of more rigorous and extensive estimation of aerial emissions. We added more elaborated calculations of aerial fluxes, taking into account the Lena River main stem (43% of the whole water area, as calculated in response to this request) and partial contribution of largest tributaries (according to their catchment areas); see our answer above.

We would like to note that while the summer period non-covered in this study (July-August and September) is characterized by slightly lower water surface areas, the water temperature and in-stream organic matter processing are higher than in spring and thus the overall $CO_2$ emissions during these months of the year might be sizably higher than those during the spring. We prefer to avoid extensive speculation on seasonaliy as it was not within the scope of the present work (which is a first snapshot assessment of C pattern in the Lena basin). However, we are confident that possible variations in water surface areas (including the contribution of very small streams, see section 4.2) do not exceed the range of uncertainties on emissions estimated in this study.

REVIEWER: Comment to lines 383-385: You compare your estimated C evasion to the DOC+DIC lateral export of the Lena River determined by other studies. Since you also collected DOC and DIC data I was wondering if how your data to compares to that of those studies?

RESPONSE: The lateral C export by the Lena River is based on regular (monthly to weekly) monitoring of dissolved C concentration and daily discharges at the terminal gauging station of Kusyur, some 700 km downstream the most northern sampling point of this study. The spatial variations of DOC and DIC concentrations obtained in this study cannot be used for calculating the lateral export. The reviewer is right when requesting to present a comparison of our data with those of other studies and we added the following text in the revised section 3.1.:

Generally, the concentrations of DOC measured in the present study during the peak of the spring flood are at the highest range of previous assessments during summer baseflow (around 5 mg L$^{-1}$; range of 2 to 12 mg L$^{-1}$, Kuzmin et al., 2009; Cauwet and Sidorov, 1996; Lara et al., 1998; Lobbes et al., 2000; Kutscher et al., 2017).

The DIC concentration in the main stem during spring flood was generally lower than that reported during summer baseflow (around 10 mg L$^{-1}$; range 5 to 50 mg L$^{-1}$) but consistent with the values reported in Yakutsk during May and June period (7 to 20 mg L$^{-1}$, Sun et al., 2018). Sizable decrease in DIC concentration between the headwaters (first 500 km of the river) and its middle course was also consistent with the Alkalinity pattern reported in previous works during summer baseflow (Pipko et al., 2010; Semiletov et al., 2011). For the Lena river tributaries, the most comprehensive data set on major ions was acquired in July-August of 1991-1996 by Huh and Edmond's group (Huh and Edmond, 1999; Huh et al., 1998a, b) and by Sun et al. (2018) in July 2012 and end of June 2013. For most tributaries, the concentration of DIC was a factor of 2 to 5 lower during spring food compared to summer baseflow. This is highly expected result given strong dilution of carbonate-rich groundwaters feeding the river in spring high flow compared to summer low flow.

REVIEWER: In L243 I understood that you compared it to your own collected DOC and DIC data, or is this also a comparison with published results? In that case a reference in L243 would be good. In general there is a lack of further discussion of your DOC and DIC data.
RESPONSE: In this part of the text, we describe the spatial variability of DOC and DIC concentrations obtained in this study. We do not extensively discuss these data because there are no sizable diurnal variations. We did examine the DOC variability in the tributaries, and, in response to other reviewers, we tested a link between DIC (and pH) and $CO_2$ concentration in the main steam and tributaries (new Fig. S7 of the Supplement).
We did not find any sizable control of these hydrochemical parameters on pCO2 in the river water. In response to this comment, extensive discussion of DOC and DIC results in comparison with literature data is presented in our response above and now included in the revised manuscript (section 3.1, L 271-284).

REVIEWER: The grammar and sentence structuring throughout needs improving, this will greatly help with the overall cohesion and readability.
RESPONSE: We agree and invested in revision of grammar and syntax of the text seeking a help of native English speaking scientist.

REVIEWER: L 327 says there was no relationship but then in brackts says: ($p < 0.05$)
RESPONSE: the last term is not needed here, revised.

We corrected inconsistent use of units as noted by reviewer in the revised version.

**Table R1.** Measured water temperature, $pCO_2$, calculated $CO_2$ flux, $CH_4$, DOC, and DIC concentrations and pH in the Lena River main stem (average ± s.d.; (n) is number of measurements). The $CO_2$ emission fluxes ($FCO_2$) are calculated for two values of transfer coefficient ($k$) of 4.464 m d$^{-1}$ (Karlsson et al., 2021) and 3.00 m d$^{-1}$ (lowerst range of world rivers in Raymond et al., 2013).

| River transect | $T_{water}$, °C | $pCO_2$, µatm | $FCO_2$, g C m$^{-2}$ d$^{-1}$ $k$ = 4.464 | $FCO_2$, g C m$^{-2}$ d$^{-1}$ $k$ = 3.00 |
|---|---|---|---|---|
| Lena upstream of Kirenga (0-578 km) | 12.65±0.22 (99) | 714±22 (99) | 0.849±0.061 (99) | 0.571±0.041 (99) |
| Lena Kirenga – Vitim (579-1132 km) | 9.17±0.15 (87) | 806±8.8 (87) | 1.19±0.024 (87) | 0.802±0.016 (87) |
| Lena Vitim -Nuya (1132-1331 km) | 8.10±0.115 (27) | 797±22 (27) | 1.22±0.072 (27) | 0.817±0.048 (27) |
| Lena Nuya – Tuolba (1331-2008 km) | 9.61±0.09 (95) | 846±12 (95) | 1.29±0.034 (95) | 0.868±0.023 (95) |
| Lena Tuolba – Aldan (2008-2381 km) | 10.6±0.21 (52) | 1003±28 (52) | 1.69±0.081 (5) | 1.21±0.048 (52) |

| | $CH_4$, µmol L$^{-1}$ | DOC, mg L$^{-1}$ | DIC, mg L$^{-1}$ | pH |
|---|---|---|---|---|
| Lena upstream of Kirenga (0-578 km) | 0.068±0.003 (6) | 13.9±1.4 (6) | 20.0±1.2 (6) | 8.12±0.203 (7) |
| Lena Kirenga – Vitim (579-1132 km) | 0.040±0.002 (12) | 7.55±0.246 (14) | 6.30±0.485 (14) | 7.77±0.040 (14) |
| Lena Vitim -Nuya (1132-1331 km) | 0.038±0.003 (5) | 9.02±0.29 (3) | 4.55±0.70 (3) | 7.69±0.063 (3) |
| Lena Nuya – Tuolba (1331-2008 km) | 0.037±0.002 (6) | 10.4±0.78 (2) | 5.09±1.157 (2) | 7.62±0.052 (2) |
| Lena Tuolba – Aldan (2008-2381 km) | 0.088±0.034 (5) | 11.6±0.27 (5) | 5.24±0.102 (5) | 7.49±0.044 (5) |

**Table R2.** Measured water temperature, $pCO_2$, calculated $CO_2$ flux, $CH_4$, DOC, DIC concentration and pH in the tributaries (average ± s.d.; (n) is number of measurements). The $CO_2$ emission fluxes ($FCO_2$) are calculated for two values of transfer coefficient ($k$) of 4.464 m d[-1] (Karlsson et al., 2021) and 3.00 m d[-1] (lowest range of world rivers in Raymond et al., 2013).

| Tributary | $T_{water}$, °C | $pCO_2$, µatm | $FCO_2$, g C m$^{-2}$ d$^{-1}$ $k$ = 4.464 | $FCO_2$, g C m$^{-2}$ d$^{-1}$ $k$ = 3.00 |
|---|---|---|---|---|
| №4 Orlinga (208 km) | 8.0±0.0 (13) | 515±2.9 (13) | 0.347±0.01 (13) | 0.233±0.005 (13) |
| №5 Nijnaya Kitima (228 km) | 6.8±0.0 (11) | 462±9.4 (11) | 0.193±0.03 (11) | 0.130±0.020 (11) |
| №8 Taiur (416 km) | 8.5±0.0 (10) | 575±31 (10) | 0.523±0.095 (10) | 0.351±0.064 (10) |
| №10 Bol. Tira (529 km) | 11.9±0.0 (15) | 788±12 (15) | 1.04±0.03 (15) | 0.701±0.021 (15) |
| №12 Kirenga (579 km) | 10.2±0.0 (323) | 448±4 (323) | 0.131±0.01 (323) | 0.088±0.008 (323) |
| №25 Thcayka (1025 km) | 8.6±0.01 (8) | 856±13 (8) | 1.37±0.04 (8) | 0.922±0.026 (8) |
| №28 Tchuya (1110 km) | 5.9±0.0 (5) | 751±5.7 (5) | 1.16±0.019 (5) | 0.779±0.013 (5) |
| №29 Vitim (1132 km) | 6.8±0.0 (10) | 654±10 (10) | 0.812±0.03 (10) | 0.602±0.018 (10) |
| №32 Ykte (1265 km) | 4.9±0.0 (11) | 676±4.8 (11) | 0.943±0.02 (11) | 0.634±0.011 (11) |
| №34 Kenek (1312 km) | 7.60±0.0 (11) | 710±2.6 (11) | 0.964±0.01 (11) | 0.648±0.005 (11) |
| №36 Nuya (1331 km) | 11.8±0.0 (10) | 752±6.0 (10) | 0.947±0.02 (10) | 0.637±0.011 (10) |
| №38 Bol. Patom (1670 km) | 6.9±0.0 (5) | 730±12 (5) | 1.05±0.04 (5) | 0.706±0.026 (5) |

| | | | |
|---|---|---|---|
| **№39 Biriuk (1712 km)** | 14.2±0.0 (5) | 929±19 (5) | 1.32±0.05 (5) | 0.888±0.032 (5) |
| **№40 Olekma (1750 km)** | 6.4±0.0 (11) | 802±14 (11) | 1.30±0.05 (11) | 0.876±0.032 (11) |
| **№43 Markha (1948 km)** | 17.5±0.0 (15) | 844±15 (15) | 0.998±0.03 (15) | 0.671±0.023 (15) |
| **№44 Tuolba (2008 km)** | 12.3±0.0 (305) | 1181±6 (305) | 2.08±0.02 (305) | 1.395±0.010 (305) |
| **№46 Siniaya (2118 km)** | 18.5±0.0 (24) | 894±19 (24) | 1.08±0.04 (24) | 0.727±0.029 (24) |
| **№48 Buotama (2170 km)** | 18.5±0.0 (24) | 1160±25 (24) | 1.66±0.06 (24) | 1.12±0.037 (24) |
| **№52-54 Aldan (2381 km)** | 14.8±0.02 (316) | 1715±12 (316) | 3.23±0.03 (316) | 2.17±0.02 (316) |

---

## Author Response (AR2)

**Dear Dr Ji-Hyung Park**

**We revised the manuscript following your suggestions, in particular:**

- Lines 35-36, 41 (tracked-changes version): It would be more helpful for readers to understand how you reached these emission estimates, if you provide your selected literature value of k, probably at the first place you mention CO2 emission estimation (L 35).
**We thank you for pointing this out and added $k = 4.46$ m d$^{-1}$ for clarity.**

- Lines 195-197: Please refer the cited and other relevant papers (e.g., Wanninkhof et al.; Lauwerwald et al.) to correct "Cwater - Cair" to "pCO2water – pCO2air".
**We agree and added a relevant sentence together with necessary reference (Lauerwald et al., 2015).**

- Line 429: suggest "that" it is; Also clarify what is "it" here?
**"…others suggest that DOM photo- and bio-degradation is low.."; revised for clarity.**

- Lines 526 & other "needle-lead deciduous trees": Do you mean "needle-leaf deciduous tress"?
**"Needle-leaf", corrected. Thanks for catching this; we are sorry for this misprint.**

- Table 3 "RPearson": Please remove a number superimposed on the title.
**Revised accordingly**

- Fig. 1: Please add the title to the vertical axis of the lower panel plot.
**This is mean multi-annual monthly discharge (Q, m$^3$ s$^{-1}$), added as necessary.**

**We thank you for handling our paper**
**Sincerely**

**Oleg S Pokrovsky**